# Insights into seasonal variation of wet deposition over Southeast Asia via precipitation adjustment from the findings of MICS-Asia III

Syuichi Itahashi[1], Baozhu Ge[2,3,4], Keiichi Sato[5], Zhe Wang[2,4,6], Junichi Kurokawa[5], Jiani Tan[7,8], Kan Huang[8,9], Joshua S. Fu[8], Xuemei Wang[10], Kazuyo Yamaji[11], Tatsuya Nagashima[12,13], Jie Li[2,3,4], Mizuo Kajino[14,13], Gregory R. Carmichael[15], Zifa Wang[2,3,4]

[1]Environmental Science Research Laboratory, Central Research Institute of Electric Power Industry (CRIEPI), Abiko, Chiba 270–1194, Japan
[2]State Key Laboratory of Atmospheric Boundary Layer Physics and Atmospheric Chemistry (LAPC), Institute of Atmospheric Physics (IAP), Chinese Academy of Sciences (CAS), Beijing 100029, China
[3]Collage of Earth Science, University of Chinese Academy of Sciences, Beijing 100049, China
[4]Center for Excellence in Urban Atmospheric Environment, Institute of Urban Environment, Chinese Academy of Sciences (CAS), Xiamen 361021, China
[5]Asia Center for Air Pollution Research (ACAP), 1182 Sowa, Nishi-ku, Niigata, Niigata 950–2144, Japan
[6]Research Institute for Applied Mechanics (RIAM), Kyushu University, Fukuoka 816-8580, Japan
[7]Multiphase Chemistry Department, Max Planck Institute for Chemistry, Mainz 55128, Germany
[8]Department of Civil and Environmental Engineering, University of Tennessee, Knoxville, TN 37996, USA
[9]Department of Environmental Science and Engineering, Fudan University, Shanghai 200433, China
[10]Institute for Environment and Climate Research, Jinan University, Guangzhou 510275, China
[11]Graduate School of Maritime Sciences, Kobe University, Kobe, Hyogo 658–0022, Japan
[12]National Institute for Environmental Studies (NIES), Tsukuba, Ibaraki 305–8506, Japan
[13]Faculty of Life and Environmental Sciences, University of Tsukuba, Tsukuba, Ibaraki 305–8572, Japan
[14]Meteorological Research Institute (MRI), Tsukuba, Ibaraki 305–0052, Japan
[15]Center for Global and Regional Environmental Research, University of Iowa, Iowa City, IA 52242, USA

*Correspondence to*: Syuichi Itahashi (isyuichi@criepi.denken.or.jp)

**Abstract.** Asia has attracted research attention because it has the highest anthropogenic emissions in the world, and the Model Inter-Comparison Study for Asia (MICS-Asia) Phase III was carried out to foster our understanding of the status of air quality over Asia. This study analyzed wet deposition in Southeast Asian countries (Myanmar, Thailand, Lao People's Democratic Republic (PDR), Cambodia, Vietnam, the Philippines, Malaysia, and Indonesia) with the aim of providing insights into the seasonal variation of wet deposition. Southeast Asia was not fully considered in MICS-Asia Phase II due to a lack of observational data; however, the analysis period of MICS-Asia III, namely, the year 2010, is covered by ground observations of the Acid Deposition Monitoring Network in East Asia (EANET), and the coordinated simulation domain was extended to cover these observation sites. The analyzed species are wet depositions of S (sulfate aerosol, sulfur dioxide ($SO_2$), and sulfuric acid ($H_2SO_4$)), N (nitrate aerosol, nitrogen monoxide (NO), nitrogen dioxide ($NO_2$), and nitric acid ($HNO_3$)), and A

(ammonium aerosol and ammonia ($NH_3$)). The wet deposition simulated with seven models driven by a unified meteorological model in MICS-Asia III was used with the ensemble approach, which effectively modulates the differences in performance among models. By comparison with EANET observations, although the seven models generally captured the wet depositions of S, N, and A, there were difficulties capturing these in some cases. Considering the model performance for ambient aerosol concentrations over Southeast Asia, this failure of models is considered to be related to the difficulty in capturing the precipitation in Southeast Asia, especially during the dry and wet seasons. Generally, meteorological field overestimated the precipitation during the dry season, which leads to the overestimation of wet deposition during this season. To overcome this, a precipitation-adjusted approach that scaled the modeled precipitation to the observed value was applied, and it was demonstrated that the model performance was improved. Satellite measurements were also used to adjust for precipitation data, which adequately accounted for the spatio-temporal precipitation patterns, especially in the dry season. As the statistical scores were mostly improved by this adjustment, the estimation of wet deposition with precipitation adjustment was considered to be superior. To utilize satellite measurements, the spatial distribution of wet deposition was revised. Based on this revision, it was found that Vietnam, Malaysia, and Indonesia were upward-corrected and Myanmar, Thailand, Lao PDR, Cambodia, and the Philippines were downward-corrected; these corrections were up to ±40%. The improved accuracy of precipitation amount was key to estimating wet deposition in this study. These results suggest that the precipitation-adjusted approach has the potential to obtain accurate estimates of wet deposition through the fusion of models and observations.

## 1 Introduction

With the recent acceleration of its emission from anthropogenic sources, Asia has the world's highest acid deposition (Vet et al., 2014). To measure atmospheric concentrations and depositions in Asia, the Acid Deposition Monitoring Network in East Asia (EANET) has maintained an observation network over Asia since 2000. At present, 13 countries participate in EANET (EANET, 2020a). This observational study is essential for understanding the status of air quality over Asian countries. Another approach is analysis based on chemical transport models (CTMs), which numerically simulate various processes of air pollutants such as emission, transport, chemical reactions, and deposition. CTMs are based on the forefront scientific algorithms; however, uncertainties in each process are critical (Carmichael et al., 2008a). Therefore, relying on a single CTM can lead to the misinterpretation of phenomena. In order to account for uncertainties in CTMs, multi-model inter-comparison study is vital. The Model Inter-Comparison Study for Asia (MICS-Asia) has been conducted over Asian countries: Phase I during 1998–2000 (Carmichael et al., 2002), Phase II during 2003–2008 (Carmichael et al., 2008b), and Phase III during 2010–2020. Phase III contains three parts: Topic 1, involving the comparison and evaluation of current air quality models (Akimoto et al., 2019, 2020; Chen et al., 2019; Itahashi et al., 2020; Kong et al., 2020; Li et al., 2019); Topic 2, involving the development of emission inventories for Asia (Li et al., 2017); and Topic 3, involving the study of interactions between air quality and

climate change (Gao et al., 2018, 2020). In terms of deposition, Itahashi et al. (2020) presented an overview of model performances in MICS-Asia III and reported that models generally captured the observed wet deposition; however, it was found that models underestimated the wet deposition of sulfate aerosol ($SO_4^{2-}$), and the differences in modeling performance were largest for nitrate aerosol ($NO_3^-$). For sulfur species, Tan et al. (2020) analyzed the oxidation ratio of sulfur (i.e., the conversion ratio from sulfur dioxide ($SO_2$) to $SO_4^{2-}$) and found that models underestimated the oxidation rate and thus underestimated the concentration and deposition of $SO_4^{2-}$. In China, which is one of the dominant anthropogenic emission sources in Asia, publicly available observational data were once quite limited (Chan and Yao, 2008). However, a nationwide estimation of nitrogen burden has been reported by Liu et al. (2013) and a national observation network has been established (see Ge et al. (2020), and references therein). The use of large amounts of observational data for China is one of the advantages of MICS-Asia III. Ge et al. (2020) analyzed the reactive nitrogen deposition over China, and the results indicated that wet deposition of ammonium aerosol ($NH_4^+$) was underestimated by all models across China.

This study focuses on Southeast Asia. This area has received research attention due to its severe air pollution, which in some cases is caused by emissions from biomass burning (Itahashi et al., 2018; Vadrevu and Justice, 2011). Recently, the 7-Southeast Asian Studies (7SEAS) program was formed to facilitate interdisciplinary research (Lin et al. 2013; Reid et al. 2013). Due to the lack of observational data from EANET, the status of deposition over Southeast Asia was not fully analyzed in Phase II of MICS-Asia. However, in Phase III, EANET observational data are available and Southeast Asian countries are fully covered by the simulation domain in CTMs. In an overview paper (Itahashi et al., 2020), we presented the acid deposition status over Asia; however, this presentation was mostly limited to the annual-accumulated status. Over Southeast Asia, which experiences distinct dry and wet seasons, wet deposition varies dramatically between these seasons. Detailed analysis is required to advance our understanding of the wet deposition status over this region, which motivated the present study. Additionally, in Itahashi et al. (2020), we reported the uncertainty of the current model-based estimation of wet deposition and proposed two approaches for improving this estimation, namely model ensemble and precipitation adjustment. The former can modulate the differences between models and the latter can adjust the precipitation amount based on observational data. A total of eight Southeast Asian countries participate in EANET. Fig. 1 shows a map of the EANET observation sites over Southeast Asia whose data were used in this study. Hereafter, Myanmar, Thailand, the Lao People's Democratic Republic (PDR), Cambodia, and Vietnam are taken to constitute continental Southeast Asia, and the Philippines, Malaysia, and Indonesia are taken to constitute oceanic Southeast Asia. The available EANET observation sites are limited over Southeast Asia; therefore, spatial interpolation methods (e.g., Kriging, land use regression) that directly use observational data (Briggs et al., 2000; Ross et al., 2007; Araki et al., 2017) may be difficult to apply. Under the framework of MICS-Asia III, an emission inventory over Asia was developed as MIX emissions (Li et al., 2017), and this is used for input data on CTMs in MICS-Asia III and subsequently conducted model inter-comparison study over Asia. Producing maps of the estimated wet deposition through CTMs can be a reasonable approach to achieve this goal. This paper is organized as follows. Section 2 describes the MICS-Asia Phase III in terms of the framework of model intercomparison and observational data. Section 3 presents the results of the analysis of the wet

depositions over Southeast Asia and discusses the problems in the current models. Section 4 explains how the precipitation-adjusted approach was applied and demonstrates that it improved the modeling performance for wet deposition. The precipitation data used to linearly scale the modeled precipitation were EANET observational data reported previously (Itahashi et al., 2020), and satellite measurements were also used in this study to improve upon this previous study. Furthermore, the wet deposition amount and the fraction of wet deposition occurring during the dry and wet seasons are presented before and after the application of the precipitation-adjusted approaches. Additionally, revised wet deposition maps over Southeast Asia are presented. Finally, Section 5 gives a summary of this study and looks toward the next Phase IV of MICS-Asia.

## 2 Framework of MICS-Asia Phase III for wet deposition

### 2.1 Model description

In MICS-Asia Phase III, the target year was 2010. The participating models were requested to submit the monthly accumulated dry and wet deposition amounts of S species ($SO_4^{2-}$, $SO_2$, sulfuric acid ($H_2SO_4$)), N species ($NO_3^-$, nitrogen monoxide (NO), nitrogen dioxide ($NO_2$), nitric acid ($HNO_3$)), and A species ($NH_4^+$ and ammonia ($NH_3$)). In total, nine models (M1, M2, M4, M5, M6, M11, M12, M13, and M14; these numbers are unified for MICS-Asia Phase III) were used in this deposition analysis; these models are summarized in an overview paper (Itahashi et al., 2020, Table 1). In this study, seven models (M1, M2, M4, M5, M6, M11, and M12) that using the same meteorological fields simulated by the Weather Research and Forecasting (WRF) model version 3.4.1 (Skamarock et al., 2008) over the unified modeling domain were selected. The unified modeling domain covered the whole of Asia with a horizontal grid resolution of 45 km and 40 vertical layers from the surface up to 10 hPa. Descriptions of the seven models are listed in Table 1. Models M1, M2, M4, M5, and M6 were from the Community Multiscale Air Quality (CMAQ) modeling system (Byun and Schere, 2006) developed by the U.S. Environmental Protection Agency (EPA), but were configured differently in terms of model version, horizontal and vertical advection/diffusion schemes, gas-phase and aerosol chemistry, dry and wet deposition schemes, and lateral boundary conditions. M11 was the nested air quality prediction model system (NAQPMS) developed by the Institute of Atmospheric Physics (IAP), Chinese Academy of Sciences (CAS) (Ge et al., 2014; Li et al., 2016), and M12 was the non-hydrostatic mesoscale model coupled with a chemistry transport model (NHM-Chem) developed by the Meteorological Research Institute (MRI), Japan (Kajino et al., 2019a). For the CMAQ models, the versions were the same for some models; however, the internal settings of advection and diffusion were different, and an aerosol scheme with thermodynamics was updated. The boundary conditions were different for the CMAQ models. The input emissions data were unified for all models using the MIX inventory (Li et al., 2017). The details of the model configurations and the verification of model performances have been published for gas (Kong et al., 2020; Li et al., 2019), aerosols (Chen et al., 2019), and deposition (Ge et al., 2020; Itahashi et al., 2020; Tan et al., 2020).

To reduce the uncertainty in various processes and configurations of the models, an ensemble approach was applied to the model results. In the findings of MICS-Asia Phase II, it was clarified that the ensemble means, rather than means of individual models, agreed well with observed sulfate and total ammonium levels (Hayami et al., 2008). In another model comparison study, namely, the Air Quality Model Evaluation International Initiative (AQMEII), which focuses on North America and Europe, model performance was improved by using the ensemble mean (Solazzo et al., 2012). In MICS-Asia Phase III, an ensemble approach for the gas species $NO_2$, $NH_3$, and CO (Kong et al., 2020), $O_3$ (Li et al., 2019), aerosols (Chen et al., 2019), and depositions (Ge et al., 2020; Itahashi et al., 2020; Tan et al., 2020) has been used and has generally performed well compared with each model. The equation used to calculate the ensemble mean (ENS) is as follows:

$$ENS = \frac{1}{N} \sum WD \qquad (1)$$

where WD is the wet deposition, and N is the number of models, which is seven in this study. Solazzo et al. (2012) proposed a method to produce a better ensemble. In the deposition analysis of MICS-Asia Phase III, a simple ensemble and a weighted ensemble were performed using the correlation coefficient (R) between the modeled and observed wet deposition (Itahashi et al., 2020). It was found that R was always improved by the weighted ensemble; however, biases can be worse in a weighted ensemble for some cases. Therefore, a simple ensemble based on the arithmetic average was applied in this study. The calculated ENS was compared with observations over Southeast Asia.

## 2.2 EANET observations

In EANET, wet deposition is observed by a wet-only sampler that is designed to collect samples during precipitation (EANET, 2010). The locations of the observation sites used in this study are plotted in Fig. 1, and Table 2 shows the latitude, longitude, altitude, sampling interval, and classification information for each. The identification numbers of the sites are unified with the overview paper of deposition analysis of MICS-Asia III (Itahashi et al., 2020). The site classification is defined as follows: urban sites are located in urbanized and industrialized areas; rural sites are located more than 20 km away from large pollution sources; and remote sites are located more than 50 km away from large pollution sources and more than 500 m away from main roads. Ion chromatography was used to analyze anions ($SO_4^{2-}$ and $NO_3^-$) and cations ($NH_4^+$). The observational data were checked by ion balance and conductivity agreement. The data completeness was determined from the duration of precipitation coverage and total precipitation amount (EANET, 2000). The sampling intervals differed from site to site, being either daily, weekly, or every 10 days (EANET, 2020b). The monthly accumulated wet deposition at each site were used for the model evaluation. For weekly or 10-day observational data, the central observation day was regarded to represent the corresponding month, and then the monthly accumulated wet deposition was calculated. Meanwhile, the model results were simply calculated from the calendar date. For the analyzed period, observational data were not available for Vientiane (No. 37; Table 2) in Lao PDR, and therefore this location was not analyzed in this study.

To evaluate the model performance compared with EANET observations, the three statistical metrics of R, normalized mean bias (NMB), and normalized mean error (NME) were used. These are defined as follows:

$$R = \frac{\sum_1^N (O_i - \bar{O})(M_i - \bar{M})}{\sqrt{\sum_1^N (O_i - \bar{O})^2} \sqrt{\sum_1^N (M_i - \bar{M})^2}} \qquad (2)$$

$$NMB = \frac{\sum_1^N (M_i - O_i)}{\sum_1^N O_i} \qquad (3)$$

$$NME = \frac{\sum_1^N |M_i - O_i|}{\sum_1^N O_i} \qquad (4)$$

where N is the total number of paired observations (O) and models (M). Additionally, the percentage of the total that fell within a factor of 2 (FAC2), within a factor of 3 (FAC3), and within a factor of 5 (FAC5) were also calculated to judge the agreement between observations and models.

## 3 Results

### 3.1 Seasonal variation of wet deposition for each country over Southeast Asia

### 3.1.1 Myanmar

Myanmar has one EANET site for wet deposition, at Yangon (No. 30; Table 2). A comparison between observational and model-simulated data for precipitation and wet depositions is shown in Fig. 2. In 2010, the observed monthly accumulated precipitation was zero from January to April, 7.5 mm in November, 25.4 mm in December, and around 300 mm from May to October. Hereafter, precipitation of 50 mm/month is used as the threshold to divide the dry and wet seasons. Based on this criterion, the dry and wet seasons were clearly characterized from observed precipitation; however, the model simulated light precipitation of around 20 mm even during the dry season, and underestimated precipitation during the wet season. Due to the seasonal variation in the observed precipitation, the observed wet depositions of S, N, and A also exhibited a clear seasonal dependency during the dry and wet seasons. Compared with the observed wet deposition, the model generally overestimated the wet deposition during the dry season and underestimated it during the wet season. At the Yangon (No. 30) site, the model variation (shown by whiskers in Fig. 2) was small for the wet depositions of S, N, and A; this indicates that the overestimation during the dry season and underestimation during the wet season was common among all models. These results indicate that the model performance for precipitation could be a critical factor in determining the model performance for wet deposition. The statistical performance of the simulated wet depositions of S, N, and A is listed in Table 3. The ENS results showed a good correlation with the observed data, with an R of around 0.8; however, there was a large underestimation for wet deposition, with an NMB greater than −70% and an NME greater than 80%. As suggested by the observed monthly wet deposition amount shown in Fig. 2, these underestimations were mainly due to the model performance during the wet season.

### 3.1.2 Thailand

In Thailand, there are six EANET sites for wet deposition, namely, Bangkok (No. 31), Samutprakarn (No. 32), Pathumthani (No. 33), Khanchanaburi (No. 34), Nakhon Ratchasima (No. 35), and Chiang Mai (No. 36; Table 2). A comparison between the observed and simulated precipitation and wet deposition is shown in Fig. 3. The dry and wet seasons were clearly distinct; the wet season is from May to October at Bangkok (No. 31), Samutprakarn (No. 32), Pathumthani (No. 33), Khanchanaburi (No. 34), and Chiang Mai (No. 36), and from March to October at Nakhon Ratchasima (No. 35). Compared to the monthly precipitation pattern, the observed monthly variations of precipitation amount and wet deposition did not show a clear relationship at Khanchanaburi (No. 34), Nakhon Ratchasima (No. 35), or Chiang Mai (No. 36). Over these sites, ambient concentrations might have contributed to the amount of the wet deposition amount. The model generally overestimated precipitation during the dry season at all six sites. For the wet depositions of S and N, the model tended to underestimate at Bangkok (No. 31), Samutprakarn (No. 32), Pathumthani (No. 33), and Nakhon Ratchasima (No. 35) during the wet season, which is related to the underestimation of precipitation itself, whereas the model overestimated precipitation at Khanchanaburi (No. 34) and Chiang Mai (No. 36) throughout the year. Large inter-model variability in the modeled wet deposition was found in some months at Khanchanaburi (No. 34). This could be related to the difference in the ambient concentration and the difference in the mechanisms of the wet deposition scheme because all models used the same meteorological field. It should be noted that all models always showed a large wet deposition in February, March, and November, despite the observed zero wet deposition amount in these months (due to the lack of precipitation during the dry season). This suggests that the discrepancy in the simulated precipitation amount could be the cause of the inaccurate simulation of wet deposition. The results of the statistical analyses are listed in Table 4. ENS showed underestimation for the wet depositions of S, N, and A, with an NMB of –20 to –50% and an NME larger than 80%. Additionally, the correlation between the observed and simulated data was small, especially for S, which showed no linear correlation. The observed wet deposition amount was higher in the wet season, but the amount modeled throughout the year was nearly constant.

### 3.1.3 Cambodia

Cambodia has one EANET site for wet deposition, at Phnom Penh (No. 38; Table 2). A comparison between the observed and simulated precipitation and wet deposition is shown in Fig. 4. The wet season (monthly accumulated precipitation more than 50 mm) lasted from March to November. According to this precipitation pattern, higher wet depositions of S, N, and A were also observed during the wet season. However, the ENS underestimated the wet deposition amount during the wet season, especially in June and July; this is related to the underestimation of precipitation in these months. All models commonly underestimated the wet deposition during the wet season. The statistical analysis is summarized in Table 5. The correlation

between the observed and simulated data was low, especially for the wet deposition of S, while the NMB and NME were around –70% and 70–80%, respectively, for the wet depositions of S, N, and A; that is, there were some difficulties in capturing the wet deposition at this site, even using the ENS.

### 3.1.4 Vietnam

Vietnam has four EANET sites for wet deposition, namely, Da Nang (No. 39), Hanoi (No. 40), Hoa Binh (No. 41), and Cuc Phuong (No. 42; Table 2). A comparison between the observed and simulated precipitation and wet deposition is shown in Fig. 5. Compared with other countries in continental Southeast Asia, precipitation patterns during the dry and wet seasons were relatively well captured at the four sites in Vietnam. Accordingly, the wet depositions of S, N, and A obtained by the ENS can generally reproduce the observed data. There were large inter-model differences when the precipitation was high. This result suggests that heavy rain events may lead to large inter-model variability in the simulated wet deposition, and the mechanisms should be further investigated. As concluded in our overview paper (Itahashi et al., 2020), this is one of the lessons learned in MICS-Asia Phase III, and this will be addressed as part of the next MICS-Asia. The results of the statistical analysis are shown in Table 6. As can be seen from the table, as well as from Fig. 5, the statistical scores for Vietnam were better than those for the other countries in continental Southeast Asia. The R value was around 0.5–0.6, while the NMB was around –35% for the wet depositions of S and N and around +15% for the wet deposition of A. The NME was around +50%, which was smaller than for other countries in continental Southeast Asia.

### 3.1.5 Philippines

There are three EANET sites for wet deposition in the Philippines, namely, Metro Manila (No. 43), Los Baños (No. 44), and Mt. Sto. Tomas (No. 45; Table 2). A comparison between the observed and simulated precipitation and wet deposition is shown in Fig. 6. The wet season was classified from June to December at Metro Manila (No. 43) and Los Baños (No. 44), and from April to November at Mt. Sto. Tomas (No. 45). Generally, the model captured the seasonal variation of precipitation adequately, but the precipitation was overestimated during the dry season. Because of this precipitation overestimation, the ENS also tended to overestimate the wet depositions of S, N, and A. Compared with other countries, the inter-model differences were larger for the sites in the Philippines. Further seeking of model wet deposition schemes focused on this region will be needed. The statistical analysis is presented in Table 7. For the wet deposition of S, R was 0.79 and NMB and NME were +11.4% and +58.0%, respectively. The ENS captured the wet deposition of S adequately. However, the NME values were worse for the wet depositions of N and A. For example, the NME for the wet deposition of A was greater than +100%. In particular, as shown in Fig. 6, the models could not reproduce the peaks of the wet depositions of N and A in October at either Metro Manila (No. 43) or Los Baños (No. 44). Additionally, the wet deposition of N and A was also underestimated for other

months during the wet season at the same two sites. This phenomenon should be further studied in the future to improve the simulation of wet deposition at these sites.

### 3.1.6 Malaysia

There are four EANET sites for wet deposition in Malaysia, namely, Petaling Jaya (No. 46), Tanah Rata (No. 47), Kuching (No. 48), and Danum Valley (No. 49; Table 2). A comparison between the observed and simulated precipitation and wet deposition is shown in Fig. 7. Compared with other countries, the four sites in Malaysia did not show clear dry and wet seasons, and precipitation amounts were consistently large over the course of the year. Therefore, the division into dry and wet seasons was not conducted for the four sites in Malaysia. At Danum Valley (No. 49), the observed precipitation was greater than 50 mm in all months except February. However, there was a lack of wet deposition observations at Danum Valley (No. 49). As shown in Fig. 7, the models had difficulties capturing the behavior of wet deposition over Malaysia. At Petaling Jaya (No. 46) and Kuching (No. 48), the ENS underestimated the wet depositions of S and N and overestimated the wet deposition of A. This tendency was common as indicated by the model-to-model variability. At these two sites, observations showed a small wet deposition of N, and the balance between cations and anions should be carefully examined. At Tanah Rata (No. 47), wet deposition was dramatically overestimated for all species. The inter-model variability was small; hence, this overestimation could be connected to the overestimation of precipitation. The results of the statistical analysis are listed in Table 8. There was a moderate correlation between the observations and simulations for the wet depositions of S and N, and the NMB and NME were highest for the wet deposition of S. It should be noted that the wet deposition of A showed much higher NMB and NME values and a lower value of R; this is due to the fact that the wet deposition of A was overestimated at all four sites in Malaysia (Fig. 7).

### 3.1.7 Indonesia

Indonesia has five EANET sites for wet deposition, namely, Jakarta (No. 51), Bandung (No. 52), Serpong (No. 53), Kototabang (No. 50), and Maros (No. 54; Table 2). A comparison between the observed and simulated precipitation and wet deposition is shown in Fig. 8. Compared with other countries in continental Southeast Asia, the dry season was shorter in Indonesia, occurring only in April in Jakarta (No. 51), Bandung (No. 52), and Serpong (No. 53), which are located on Java Island; in August in Kototabang (No. 50), which is located on Sumatra Island; and in January and February in Maros (No. 54), which is located on Sulawesi Island. The observed wet depositions of S, N, and A in these limited dry seasons were generally lower than during the wet season; however, no difference in the simulated wet depositions of S, N, and A was observed between the wet and dry seasons. The reason for this failure was that the model did not reproduce the observed reduced precipitation amounts during the dry seasons. As was found in the Philippines, the inter-model variation was large, except for Maros (No.

54), and further study focusing on this region will also be required. The results of the statistical analysis are listed in Table 9. A moderate correlation between observations and simulations was found for the wet depositions of S, N, and A, but the ENS overestimated the wet depositions of S, N, and A, especially for S, with an NMB of +65.6% and an NME larger than 100%.

## 4 Discussion

### 4.1 Proposal of precipitation-adjusted approach over Southeast Asia

As presented in Section 3, although the model performances in MICS-Asia III based on an ensemble approach generally captured the observed wet deposition over Southeast Asia, there were some difficulties in capturing the observed values. The errors in the simulated values of wet deposition are associated with ambient concentration and/or precipitation. Our previous overview paper (Itahashi et al., 2020) presented two approaches for improving the modeling of wet deposition, namely, the ensemble approach and the precipitation-adjusted approach. The former approach was used in this study. In terms of the modeling performance for the ambient concentrations of aerosols of $SO_4^{2-}$, $NO_3^-$, and $NH_4^+$, our companion paper reported better performance over Southeast Asia compared with North and East Asia (Chen et al., 2019). As noted in Section 3, the model generally overestimated precipitation as well as wet deposition during the dry season. Additionally, the model sometimes simulated non-zero precipitation, and consequently non-zero wet deposition, despite the absence of wet deposition due to the absence of precipitation. Based on these findings in MICS-Asia III, the difficulty stemmed from the inaccuracy of the modeled precipitation, which is fundamentally important for simulating the wet deposition. The precipitation-adjustment method is expected to improve model performance. The precipitation-adjusted approach linearly scales the precipitation to obtain the precipitation-adjusted wet deposition via the following equation:

$$Adjusted\ WD = \sum_{monthly} Original\ WD_{model} \times \frac{\sum_{monthly} P_{observation}}{\sum_{monthly} P_{model}} \tag{5}$$

where $WD_{model}$ is the original model-simulated wet deposition, and $P_{model}$ and $P_{observation}$ are the modeled and observed precipitation, respectively. This method involves adjusting the precipitation amount which affects the wet deposition amount on a monthly time scale. Here, it is assumed that the errors in the modeled precipitation are linearly associated with the errors in the modeled wet deposition. This approach has been used in previous studies in the U.S.A. (Appel et al., 2011; Zhang et al., 2018) and East Asia (Itahashi, 2018; Saya et al., 2018). Following our previous work in MICS-Asia III for deposition (Itahashi et al., 2020), wet depositions were adjusted on a monthly time scale and then the annual wet deposition was recalculated using the precipitation-adjusted monthly wet deposition. Adjustment using shorter time scales is difficult because the modeled precipitation ($P_{model}$ in Eq. (5)) approaches zero, which leads to unreasonably large values, and vice versa for larger time scales. The precipitation-adjusted approach using EANET observational data is hereafter called AO (adjusted by observation at EANET site).

The precipitation-adjusted approach was shown to be effective for improving the modeling reproducibility in MICS-Asia III (Itahashi et al., 2020). However, this approach has a limitation in that the adjusted wet deposition was obtained only at locations corresponding to EANET observation sites, and hence the adjusted wet deposition was spatially limited. To overcome this limitation, in this study, we additionally used a satellite dataset; this precipitation-adjusted approach is hereafter called AS (adjusted by satellite measurement). For this purpose, the Tropical Rainfall Measuring Mission (TRMM) multi-satellite precipitation analysis (TMPA) dataset was applied (Huffman et al., 2007). The used product is the latest version 7 of the 3B43 dataset, which provides monthly precipitation with the most accurate precipitation estimate covering 50° S to 50° N (TRMM, 2011). The gridded data of 0.25×0.25° were converted into the simulation domain used in MICS-Asia Phase III. For the AO and AS approaches, wet deposition in each of the seven models was first adjusted on a monthly time scale, and then the ENS was calculated using Eq. (1).

A comparison among the WRF simulation, EANET surface observations, and TRMM satellite measurements is given in Fig. 9. The results of the statistical analysis are also shown in this figure. The comparison between the EANET surface observations and the WRF model simulations showed that the model generally reproduced the observed monthly precipitation adequately, with an R of 0.56, an NMB of +24.2%, and an NME of +64.7%. However, as shown in Figs. 2–8, the model tended to overestimate low precipitation levels (see Fig. 9 for the dry season). As has been discussed for Figs. 2–8, this overestimation may be the reason for the mismatch between the simulated and observed the wet depositions of S, N, and A. Resolving this problem is important for improving simulations of wet deposition. Meanwhile, in the comparison between the EANET surface observations and the satellite measurements, the statistical scores were superior to those obtained between the modeled and observed data, with an R of 0.77, an NMB of +5.9%, and an NME of +39.5%. The correspondence between EANET surface observations and satellite measurements was better (relative to the correspondence between the modeled and observed data) for monthly precipitation of less than 50 mm. From this result, it is expected that precipitation-adjustment based on satellite measurements also has the potential to improve the original simulation of wet deposition. It should be noted that even though satellite and ground-based observations showed differences in the precipitation amount, this result indicates that further consideration of the how well precipitation is represented by the spatial resolution (broader observation by satellites and point-specific observations using ground-based monitoring) is important. Accordingly, the effect of the modeling spatial resolution on the simulated precipitation should be considered in future studies. The spatial distributions of precipitation from the WRF simulation and TRMM satellite measurements are respectively presented in Supplemental Figs. 1 and 2, and the adjustment factors for each month are given in Supplemental Fig. 3.

## 4.2 Improvements of wet deposition modeling through a precipitation-adjusted approach for each country in Southeast Asia

### 4.2.1 Myanmar

At the Yangon (No. 30) site in Myanmar, the wet depositions of S, N, and A was underestimated, with an NMB exceeding –70%, as listed in Table 3. Table 3 also provides the results of the statistical analysis for the AO and AS approaches, demonstrating that the underestimation in the ENS was improved by both approaches; most of the statistical scores were improved compared with the ENS, though there was still underestimation compared with the observed wet depositions of S, N, and A. Fig. 10 shows the annual accumulated wet depositions of S, N, and A from the observational data, ENS, AO, and AS. As shown in the figure, the wet deposition was higher with the AO and AS approaches compared with ENS; that is, the underestimation was partly improved. Fig. 10 also shows the fractions of wet deposition occurring during the dry and wet seasons as bar graphs for the observational data, ENS, AO, and AS. It can be clearly seen that, for the wet depositions of S, N, and A, the fraction during the dry season was overestimated with ENS but was well matched with the AO and AS approaches.

### 4.2.2 Thailand

The wet depositions of S, N, and A was generally underestimated at the six sites in Thailand, as shown in Table 4. The statistical scores for AO and AS are also provided in this table. For the R value and the NME, AO and AS obtained superior values for Thailand compared with the ENS, showing a stronger correlation with the observational data. For AO and AS, the R values ranged from 0.61 to 0.85 for the wet depositions of S, N, and A, and the NMB was improved by 20–30% compared with the ENS. Fig. 11 shows the annual accumulated wet depositions of S, N, and A from observational data, ENS, AO, and AS, and the fractions of deposition occurring in the dry and wet seasons. From this figure, it can be seen that, compared with the ENS, the AO and AS approaches obtained superior values of the fractions of wet deposition during the dry and wet seasons at all six sites in Thailand. For Bangkok (No. 31), Samutprakarn (No. 32), and Pathumthani (No. 33), the underestimation in ENS was improved and the annual accumulated wet depositions of S, N, and A was close to the observed value for both AO and AS. Meanwhile, at Khanchanaburi (No. 34) and Chiang Mai (No. 36), the overestimation in ENS was improved and the annual accumulated wet depositions of S, N, and A was close to the observed value for both AO and AS. These results clarify that the precipitation-adjusted approach was effective to solve both overestimation and underestimation problems in the original simulated wet deposition. However, it should be noted that, for Nakhon Ratchasima (No. 35), although the fractions of wet deposition occurring during the dry and wet seasons were improved with the AO and AS approaches, underestimation was worsened.

### 4.2.3 Cambodia

At Phnom Penh (No. 38) in Cambodia, there were some difficulties capturing the wet depositions of S, N, and A using the ENS. As shown in Table 5, there was a low correlation between the observed values and the ENS for the wet deposition of S, and an even lower correlation for the wet depositions of N and A. The NMB was around –70% and the NME was 70–80% for the wet depositions of S, N, and A using the ENS. These deficiencies in the ENS were adequately improved using AO and AS. For AO and AS, all statistical scores showed an improvement compared with the ENS. Fig. 12 shows the annual accumulated wet depositions of S, N, and A from observational data, ENS, AO, and AS, and the fraction of deposition occurring in the dry and wet seasons at Phnom Penh. It was also found that the ENS mismatched the fraction of wet deposition compared with the observed value, whereas AO and AS obtained more accurate fractions, as well as more accurate values of the annual accumulated wet deposition.

### 4.2.4 Vietnam

For the four EANET sites in Vietnam, the statistical scores for the ENS were superior to those for other countries in continental Southeast Asia. In most cases, for AO and AS, the scores were improved compared with the ENS for the wet depositions of S, N, and A, as shown in Table 6. Fig. 13 shows the annual accumulated wet depositions of S, N, and A from observational data, ENS, AO, and AS, and the fraction of wet deposition occurring in the dry and wet seasons in Vietnam. Compared with other countries in continental Southeast Asia, the fraction of wet deposition occurring during the dry and wet seasons was better predicted by the ENS, and AO and AS performed similarly. However, for AS, the fraction during the wet season was overestimated at Hoa Binh (No. 41) and underestimated at Cuc Phuong (No. 42). Additionally, the overestimated wet deposition amount of S, N, and A at Da Nang (No. 39) led to a discrepancy with the observed results.

### 4.2.5 Philippines

For the three EANET sites in the Philippines, it was found that the model overestimated the precipitation during the dry season. Fig. 14 presents the annual accumulated wet depositions of S, N, and A for the observational data, ENS, AO, and AS, and the fraction of the wet deposition occurring in the dry and wet seasons in the Philippines. As shown in the figure, the ENS overestimated the fraction during the wet season at all sites for the wet depositions of S, N, and A. However, with AO and AS, this overestimation was improved and the simulated values were close to the observed ones. The statistical scores are listed in Table 7. As shown in the table, R was not changed or slightly increased and NME was improved, but NMB was not improved. As shown in Fig. 14, this result was related to the change in model performance at Metro Manila (No. 43); the annual accumulated wet deposition amounts of S, N, and A were markedly decreased and very different from the observed data.

#### 4.2.6 Malaysia

At the four EANET sites in Malaysia, no distinction was found between the dry and wet seasons. Fig. 15 shows the annual accumulated wet depositions of S, N, and A from observation, ENS, AO, and AS, while Table 8 lists the statistical scores. AO and AS generally obtained improved results compared with the ENS. In particular, the strikingly large overestimation of the wet deposition of A in the ENS (NMB and NME greater than 200%) was improved with AO and AS. At Petaling Jaya (No. 46) and Tanah Rata (No. 47), the observed annual accumulated wet deposition of A was around 2000 g N ha$^{-1}$, whereas the ENS value was nearly 8000 g N ha$^{-1}$. This large overestimation was reduced by AO and AS, which obtained values close to the observed value.

#### 4.2.7 Indonesia

In Indonesia, during the short dry season, wet deposition showed a steep decline; however, models did not show such a dramatic decrease. As shown in Table 9, the statistical scores for AO and AS were mostly superior to those of the ENS; the moderate correlation found for the wet depositions of S, N, and A in the ENS were improved by AO and AS. For the wet deposition of S, the NMB of +65.6% and NME of +100.2% in the ENS were improved by AO and AS. Fig. 16 shows the annual accumulated wet depositions of S, N, and A from observational data, ENS, AO, and AS, and the fraction of wet deposition occurring in the dry and wet seasons in Indonesia. The overestimation of the fraction during the wet season obtained by the ENS was improved by AO, but there was no change with AS. Although the annual accumulated wet depositions of S, N, and A for the ENS were generally close to the observed values, AS showed further overestimation at Serpong (No. 53) and there was almost no change at Jakarta (No. 51).

### 4.3 Revision of the distribution of wet deposition over Southeast Asia

Based on the analysis and statistical results of the precipitation-adjusted approaches using surface observations and satellite measurements, it was found that these approaches improved the simulation of the wet deposition amount, as well as the fraction of wet deposition occurring during the dry and wet seasons. Although there were still difficulties in some cases, the precipitation adjustment was shown to be an effective way to improve the simulated wet deposition. One of the advantages of the adjustment using satellite measurements is that it provides the spatial distribution of adjustment factors; hence, it is possible to revise the wet deposition mapping over the modeling domain. In Fig. 17, the annual accumulated wet depositions of S, N, and A are mapped. Both the ENS and AS simulated hot spots with high depositions of S, N, and A in regions such as northern Vietnam, the southern Malay Peninsula, and Sumatra Island and Java Island in Indonesia. However, there were clear differences between AS and ENS. These differences were similar for the wet depositions of S, N, and A. As shown in Fig. 17

(right), for ENS, higher values (blue color) compared with AS occurred in the central part of continental Southeast Asia, such as Eastern Myanmar; Thailand; the western edge of Sumatra Island, the south of Java Island, and Sulawesi Island in Indonesia, the Philippines; meanwhile, ENS produced lower values (red color) compared with AS over northern Vietnam, the east of Sumatra Island, and the northern edge of Java Island and Kalimantan Island in Indonesia.

Finally, Fig. 18 shows the original and revised wet deposition amounts in the eight countries participating in EANET. This figure summarizes the annual accumulated wet depositions of S, N, and A by the country-scale summed amount. As can be seen from the differences between ENS and AS shown in Fig. 17, the revisions by AS were similar for the wet depositions of S, N, and A. For AS, over Vietnam, Malaysia, and Indonesia, the country-level wet depositions were revised upward, whereas they were revised downward in the other five countries. The magnitudes of these revisions were up to ±40%. The revision of wet deposition by a precipitation-adjusted approach was critically needed for the accurate estimation of wet deposition. The results of this study suggest that an approach which applies the precipitation obtained from satellite measurements could be used as one of the methodologies in the Measurement–Model Fusion for Global Total Atmospheric Deposition (MMF-GTAD) project under the Global Atmosphere Watch (GAW) program of the World Meteorological Organization (WMO) (WMO GAW, 2017, 2019). In this study, we were able to revise the wet deposition mapping over Southeast Asia to achieve better modeling reproducibility compared with EANET.

## 5 Conclusion

MICS-Asia Phase III has been conducted to understand the current modeling capabilities for wet deposition and comprehend air pollution in Asia. This study presented a detailed analysis over Southeast Asia. The ensemble means of the modeled wet depositions of S, N, and A from seven models were evaluated by comparison with the wet deposition observed by EANET. Generally, the ensemble model could capture the observed wet deposition; however, sometimes failed to capture the wet deposition and obtained low correlations and/or large biases and errors. Based on a detailed analysis of the observed precipitation at each EANET observation site, it was found that this failure to capture the wet deposition was related to the poor representation of the precipitation amount. In some cases, the model did not adequately simulate the precipitation pattern during the dry and wet seasons.

To overcome this modeling difficulty for precipitation, in this study, two precipitation-adjusted approaches were applied using EANET surface observations and TRMM satellite measurements, respectively. Both approaches have been shown to be effective for improving the modeling of the wet depositions of S, N, and A. To use satellite measurements of precipitation, the spatial mappings of wet depositions were further revised. It was found that the original modeled wet deposition was overestimated over the central part of continental Southeast Asia, the western edge of Sumatra Island, the south of Java Island and Sulawesi Island in Indonesia, and the Philippines, and was underestimated over northern Vietnam, the east of Sumatra

Island and the northern edge of Java Island and Kalimantan Island in Indonesia. For the country-scale accumulation of wet depositions, the wet deposition amounts were revised by up to ±40% by the precipitation-adjusted approaches. Similar differences were found for wet depositions of S, N, and A; upward corrections were required for Vietnam, Malaysia, and Indonesia, whereas downward corrections were required for Myanmar, Thailand, Lao PDR, Cambodia, and the Philippines. The use of meteorological models could cause large errors related to precipitation patterns, as found in this study, and the use of meteorological model ensembles could be a possible way to obtain more accurate air quality model simulations (e.g., Kajino et al., 2019b). The precipitation-adjustment approach was effective at most sites; however, no improvement was found at other sites. The understanding of the mechanisms of the wet deposition process itself should be further investigated and inter-compared in the future Phase IV. This adjustment approach might be difficult to apply at time scales shorter than one month; therefore, the performance of meteorological models for precipitation simulation should be paid further attention in order to improve the simulation accuracy of wet deposition. Additionally, greater inter-model variation was noted in the Philippines and Indonesia, especially during months with heavy precipitation. To investigate the differences on model wet deposition scheme, such heavy rainy events with finer spatio-temporal resolution should be pursued in the future MICS-Asia Phase IV.

**Data availability**

The EANET wet deposition data used in this study are available at: https://monitoring.eanet.asia/document/public/index. The TRMM satellite precipitation measurements were downloaded from: https://doi.org/10.5067/TRMM/TMPA/MONTH/7. The model results of MICS-Asia Phase III are available upon request.

**Supplement**

The supplement related to this article is available online at https://doi.org/10.5194/acp-XX-XXXX-XXXX-supplement.

**Author contributions**

SI led the deposition analysis group in MICS-Asia III, performed one of the model simulations, and prepared the manuscript with contributions from all co-authors. BG and KS are members of the deposition analysis group in MICS-Asia III and discussed the results with SI. ZW conducted the meteorological simulation driving the CTM simulation. JK prepared the emission inventory data for Southeast Asia and discussed the results from the viewpoint of emissions. TJ, JSF, XW, KY, TN,

JL, BG, and MK performed the model simulations and contributed to submit their simulated deposition results. MICS-Asia III was coordinated by GRC and ZW.

**Competing interests**

5 The authors declare that they have no conflict of interest.

**Acknowledgements**

The authors thank EANET for providing the wet deposition observational data. The authors are grateful for the satellite measurement dataset from TRMM.

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

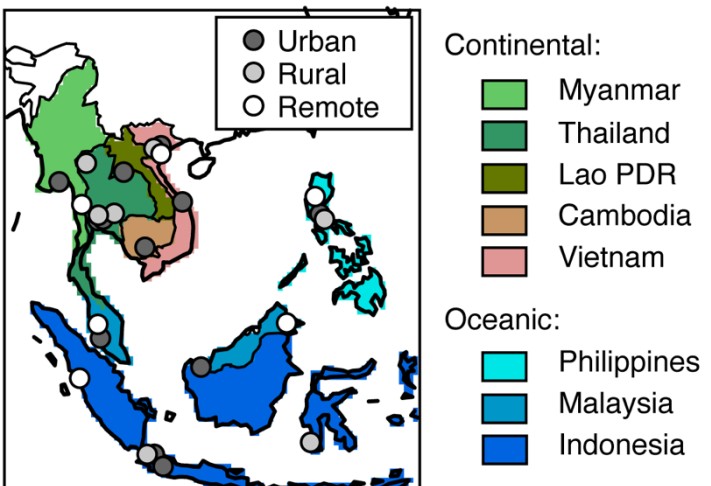

**Figure 1: Map of Southeast Asia. Circles with different colors indicate observation sites classified as remote (white), rural (light gray), and urban (dark gray) by the Acid Deposition Monitoring Network in East Asia (EANET). Map colors indicate the eight countries participating in EANET in 2010. PDR, People's Democratic Republic.**

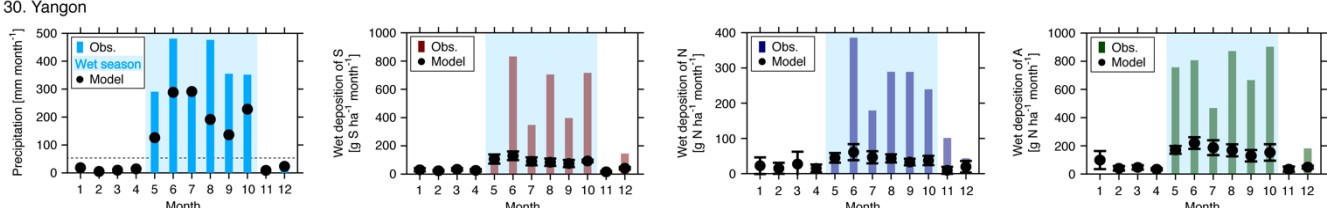

**Figure 2: Monthly accumulated precipitation and wet depositions of S, N, and A at Yangon, Myanmar. Whiskers represent the standard deviation among the seven models, and the wet season (light blue color) is defined as months when precipitation exceeded 50 mm.**

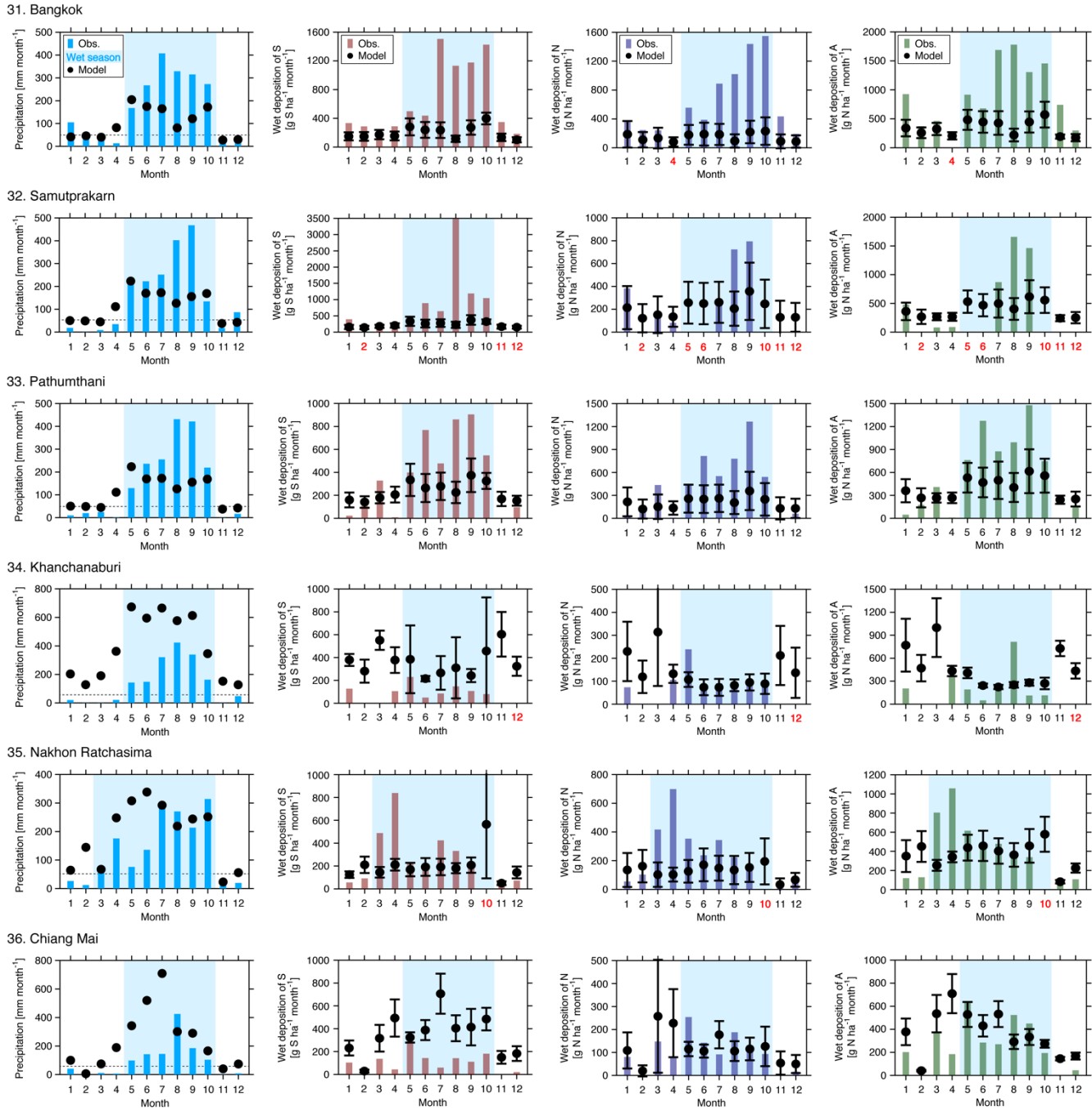

**Figure 3: Monthly accumulated precipitation and wet depositions of S, N, and A over Thailand. Whiskers represent the standard deviation among the seven models, and the wet season (light blue color) is defined as months when precipitation exceeded 50 mm. Months shown in red indicate a lack of observational data.**

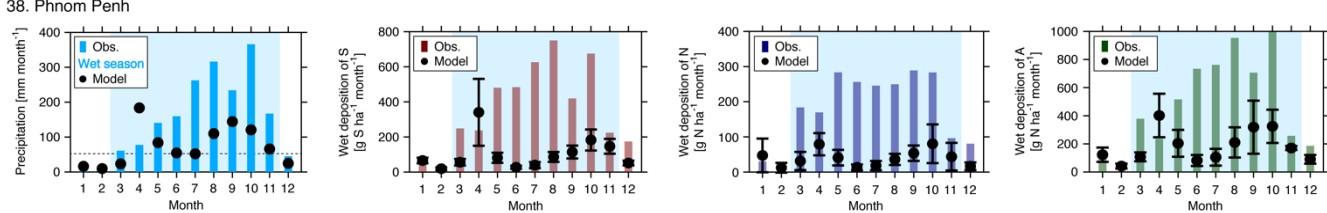

**Figure 4: Monthly accumulated precipitation and wet depositions of S, N, and A at Phnom Penh, Cambodia. Whiskers represent the standard deviation among the seven models, and the wet season (light blue color) is defined as months when precipitation exceeded 50 mm.**

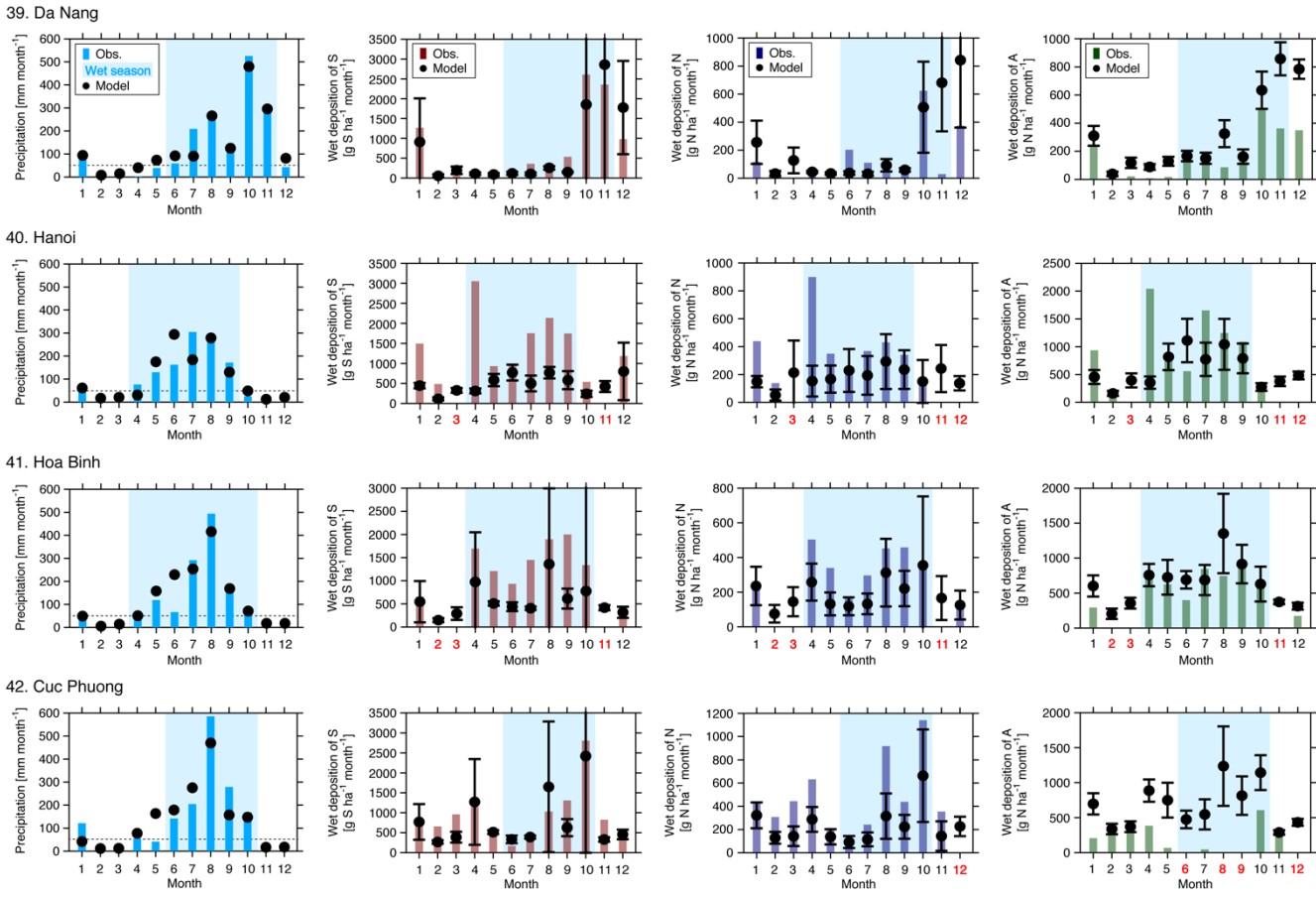

**Figure 5: Monthly accumulated precipitation and wet depositions of S, N, and A over Vietnam. Whiskers represent the standard deviation among the seven models, and the wet season (light blue color) is defined as months when precipitation exceeded 50 mm. Months shown in red indicate a lack of observational data.**

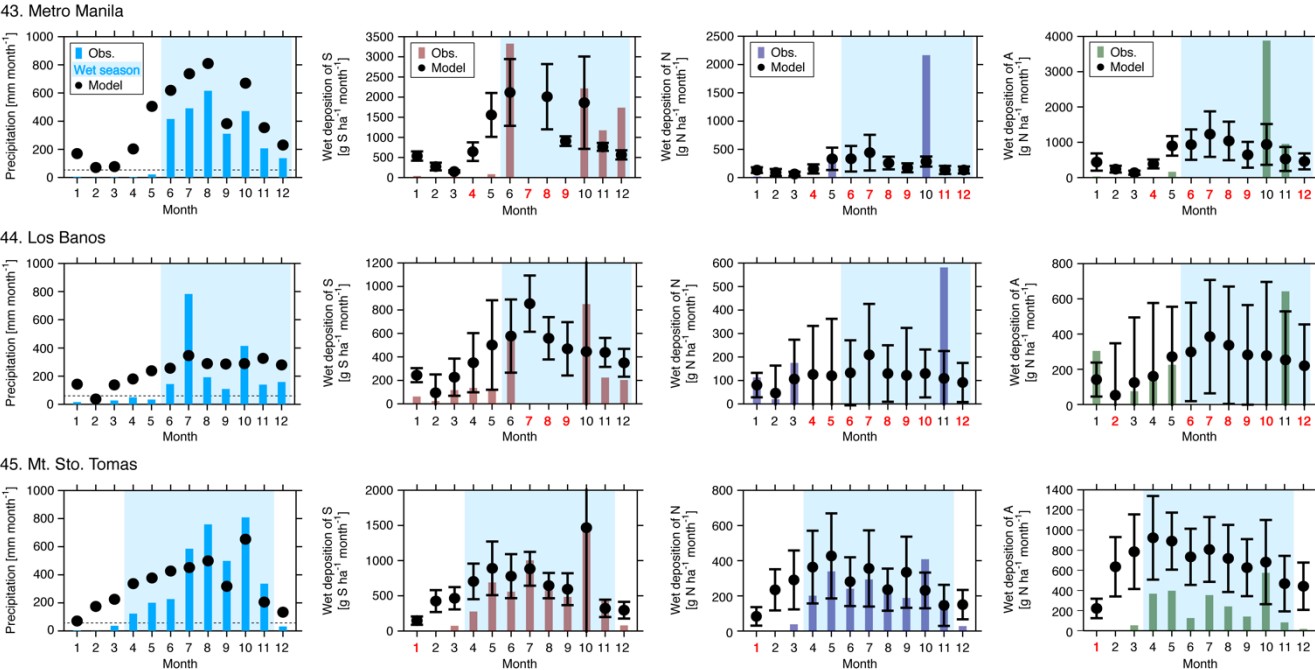

**Figure 6: Monthly accumulated precipitation and wet depositions of S, N, and A over the Philippines. Whiskers represent the standard deviation among the seven models, and the wet season (light blue color) is defined as months when precipitation exceeded 50 mm. Months shown in red indicate a lack of observational data.**

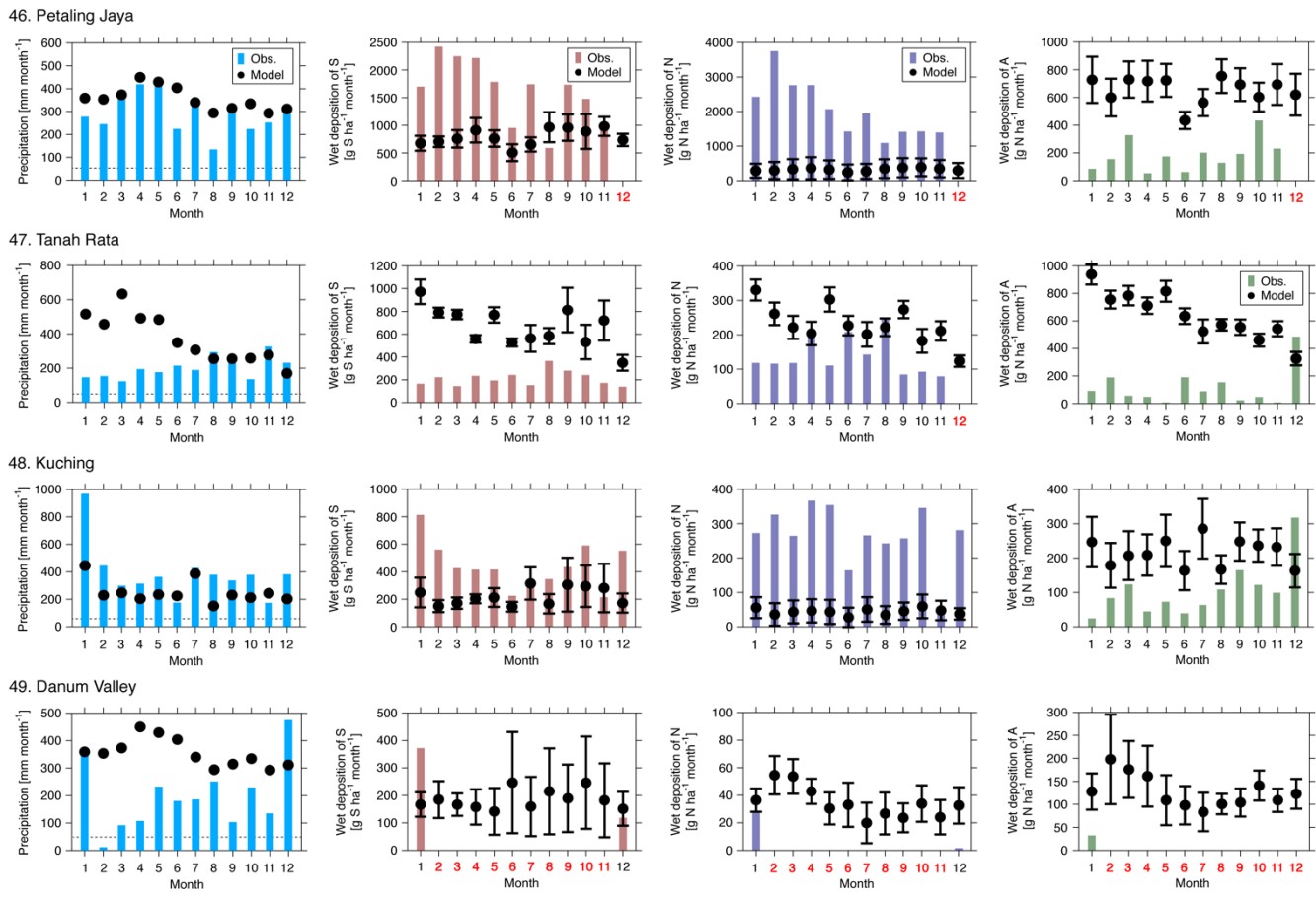

**Figure 7: Monthly accumulated precipitation and wet depositions of S, N, and A over Malaysia. Whiskers represent the standard deviation among the seven models. Months shown in red indicate a lack of observational data.**

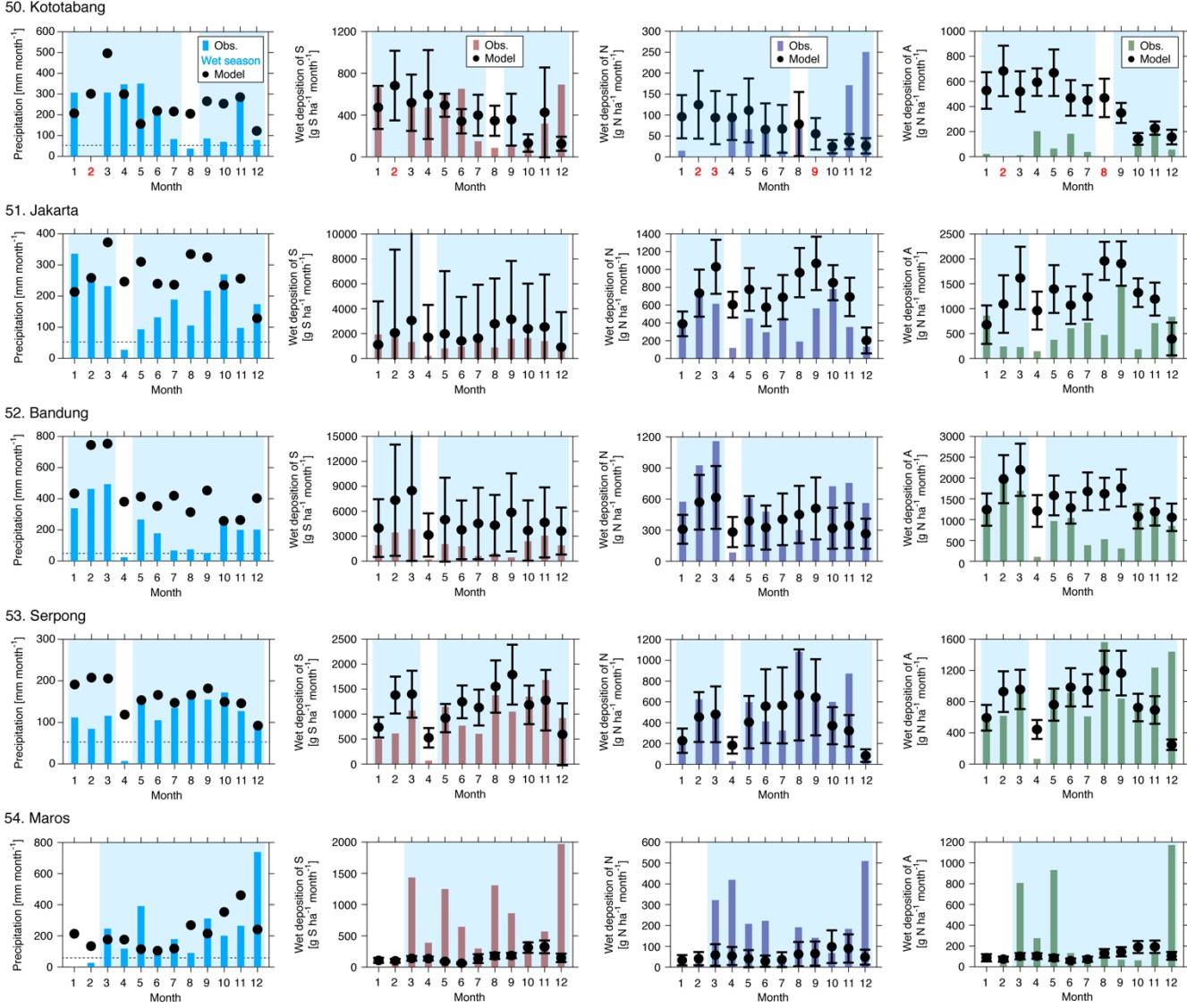

**Figure 8: Monthly accumulated precipitation and wet depositions of S, N, and A over Indonesia. Whiskers represent the standard deviation among the seven models, and the wet season (light blue color) is defined as months when precipitation exceeded 50 mm. Months shown in red indicate a lack of observational data.**

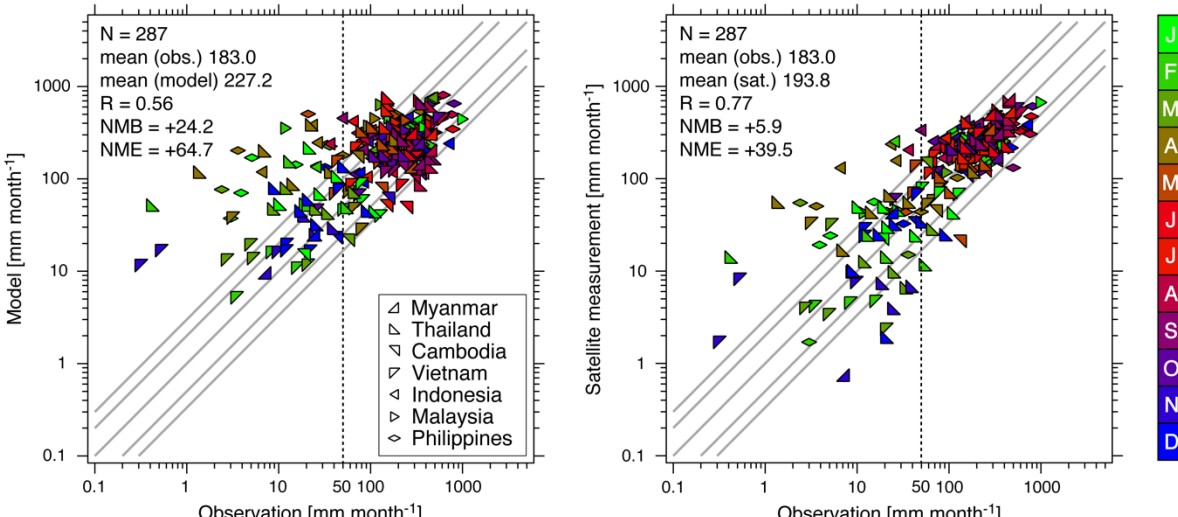

**Figure 9: Scatter plots of the monthly precipitation amount over Southeast Asia comparing EANET surface observations with (left) model simulations and (right) satellite measurements. Symbols indicate different countries and colors indicate different months. In the inset, the statistical metrics of mean, correlation coefficient (R), normalized mean bias (NMB), and normalized mean error (NME) are shown. The vertical dotted line represents observed precipitation of 50 mm month[-1], which defines the boundary between the dry and wet seasons in this study.**

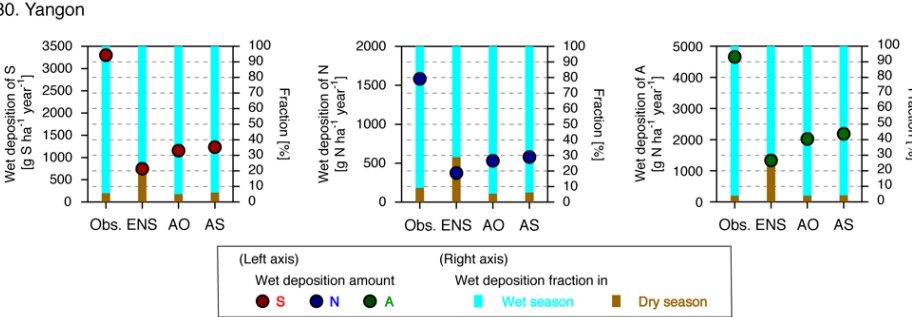

**Figure 10: Observed and simulated annual accumulated wet deposition amounts of S, N, and A, and the fraction of wet deposition during the wet and dry seasons at Yangon, Myanmar. ENS, AO, and AS stand for the results of ensemble mean, precipitation adjustment by EANET observations, and precipitation adjustment by satellite observations, respectively.**

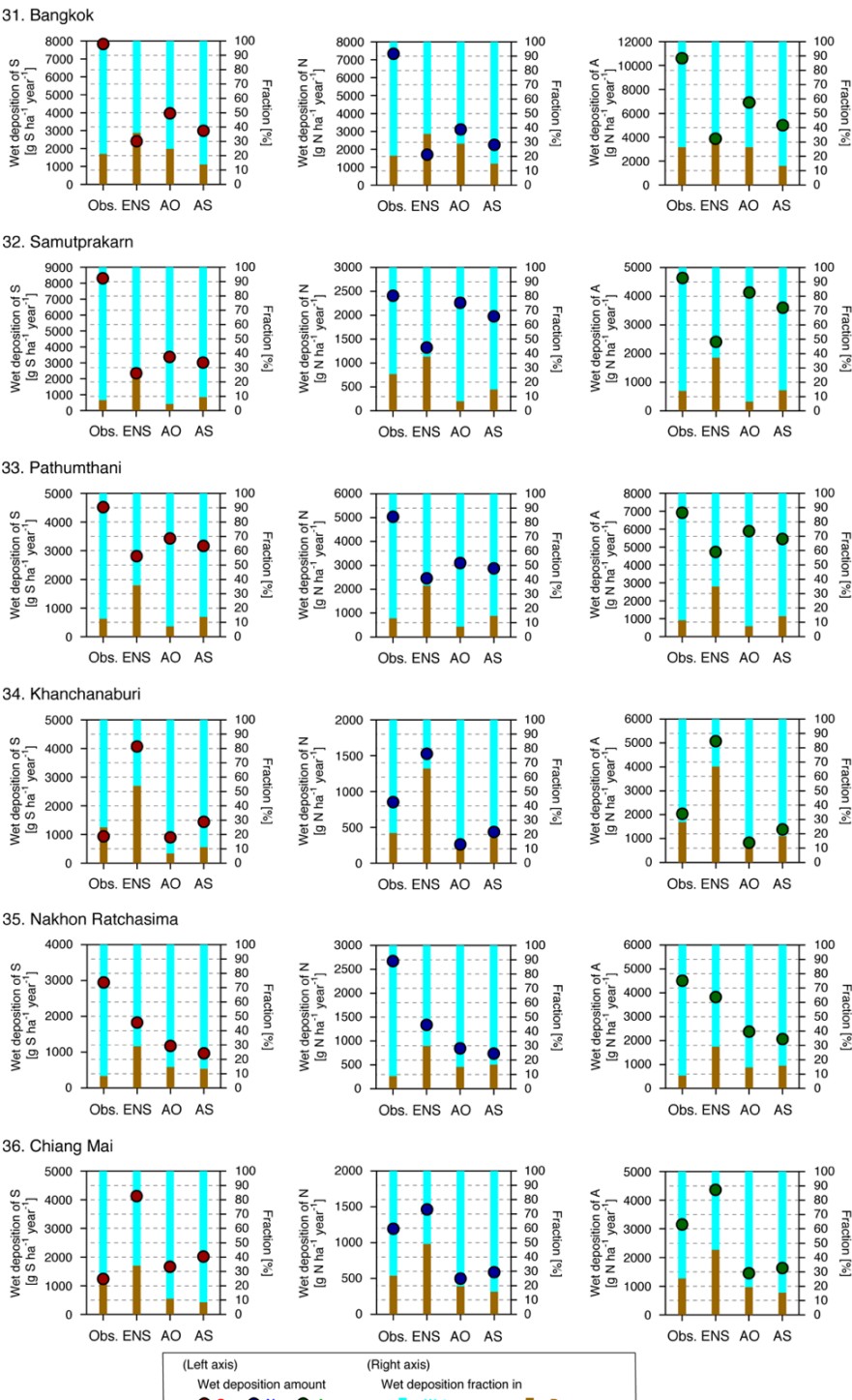

31. Bangkok

32. Samutprakarn

33. Pathumthani

34. Khanchanaburi

35. Nakhon Ratchasima

36. Chiang Mai

(Left axis)
Wet deposition amount
● S  ● N  ● A

(Right axis)
Wet deposition fraction in
■ Wet season  ■ Dry season

**Figure 11: Observed and simulated annual accumulated wet deposition amounts of S, N, and A, and the fraction of wet deposition during the wet and dry seasons at six sites in Thailand. The annual accumulated wet deposition amount is based on the months in which wet deposition observations were available (see Fig. 3).**

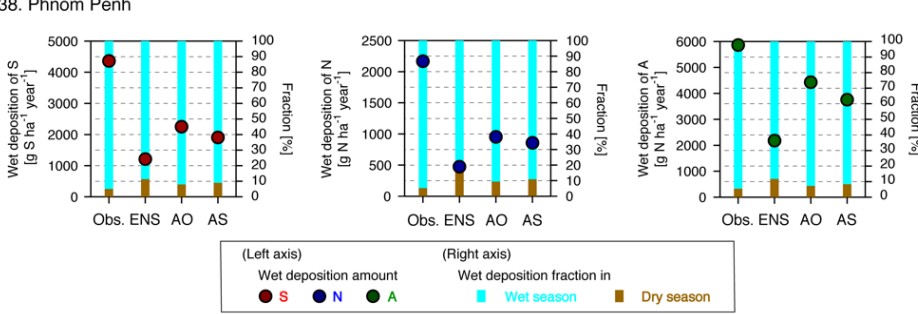

**Figure 12: Observed and simulated annual accumulated wet deposition amounts of S, N, and A, and the fraction of wet deposition during the wet and dry seasons at Phnom Penh, Cambodia.**

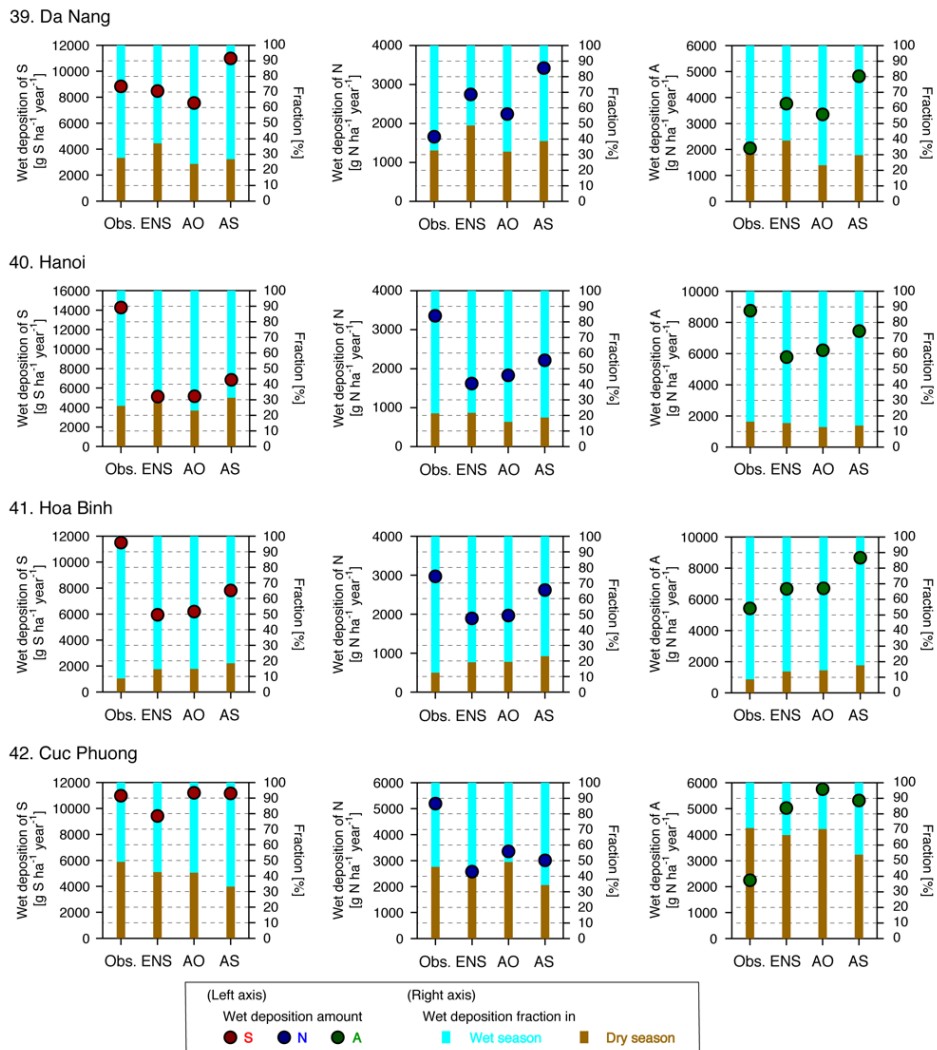

**Figure 13: Observed and simulated annual accumulated wet deposition amounts of S, N, and A, and the fraction of wet deposition during the wet and dry seasons at four sites over Vietnam. The annual accumulated wet deposition amount is based on the months for which wet deposition observations were available (see Fig. 5).**

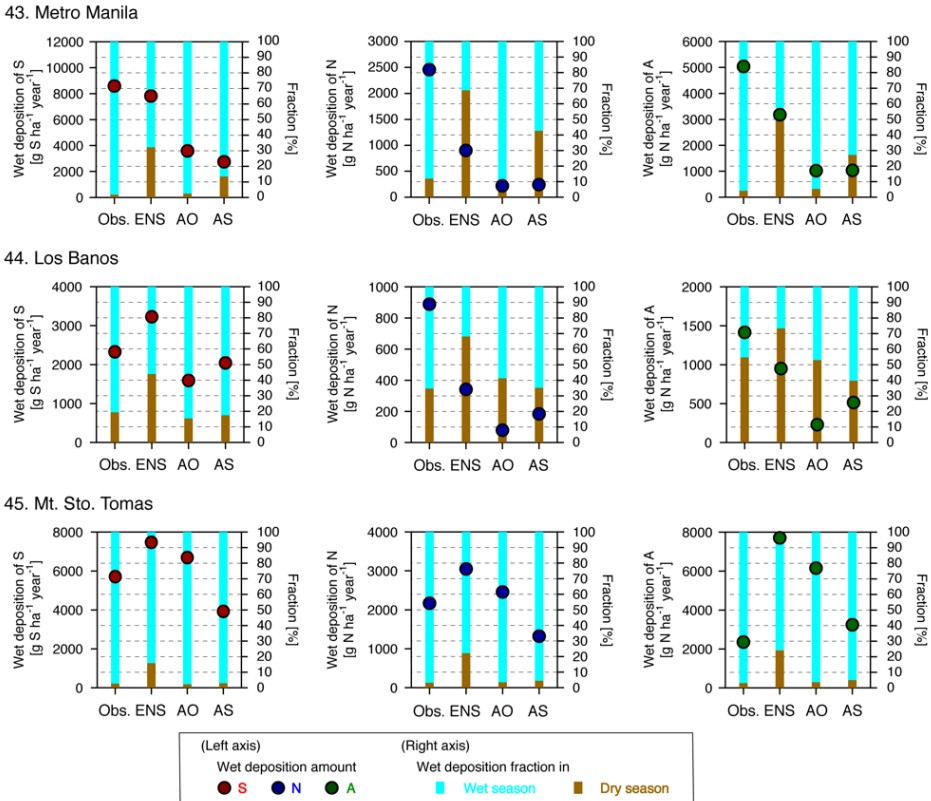

**Figure 14: Observed and simulated annual accumulated wet deposition amounts of S, N, and A, and the fraction of wet deposition during the wet and dry seasons at three sites in the Philippines. The annual accumulated wet deposition amount is based on the months for which wet deposition observations were available (see Fig. 6).**

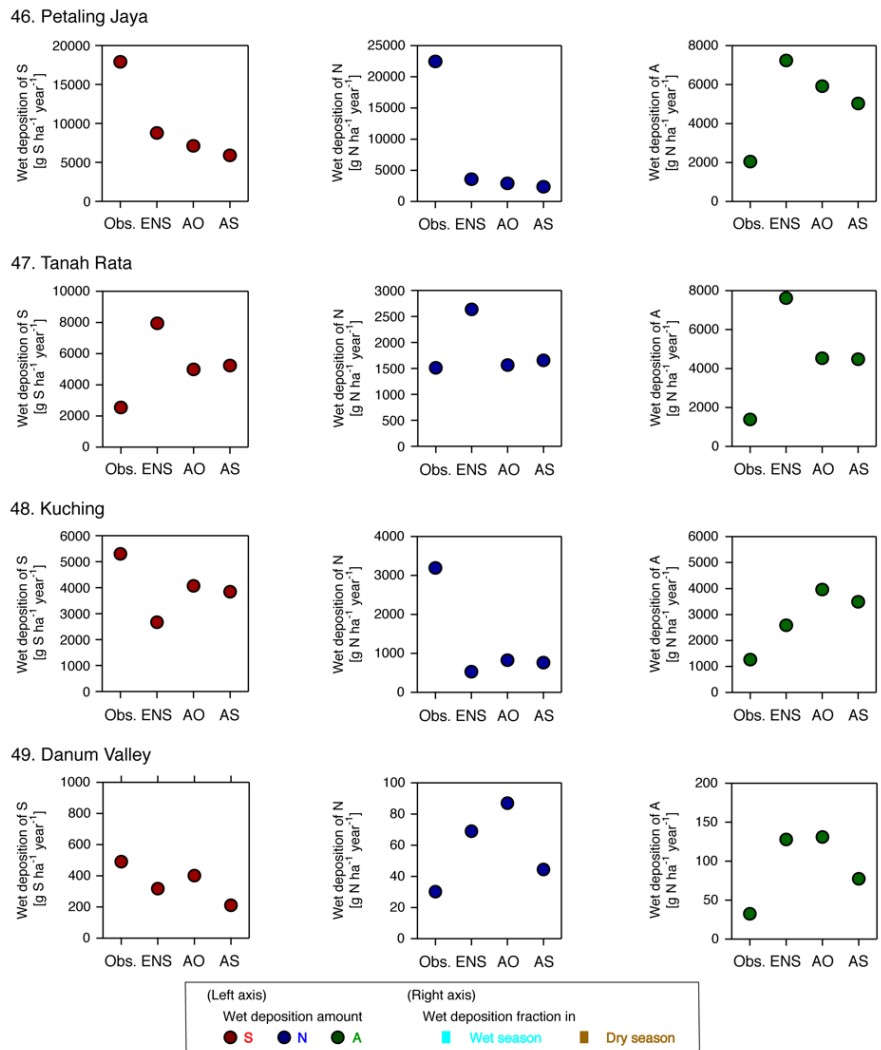

**Figure 15: Observed and simulated annual accumulated wet deposition amounts of S, N, and A at four sites in Malaysia. The annual accumulated wet deposition amount is based on the months for which wet deposition observations were available (see Fig. 7).**

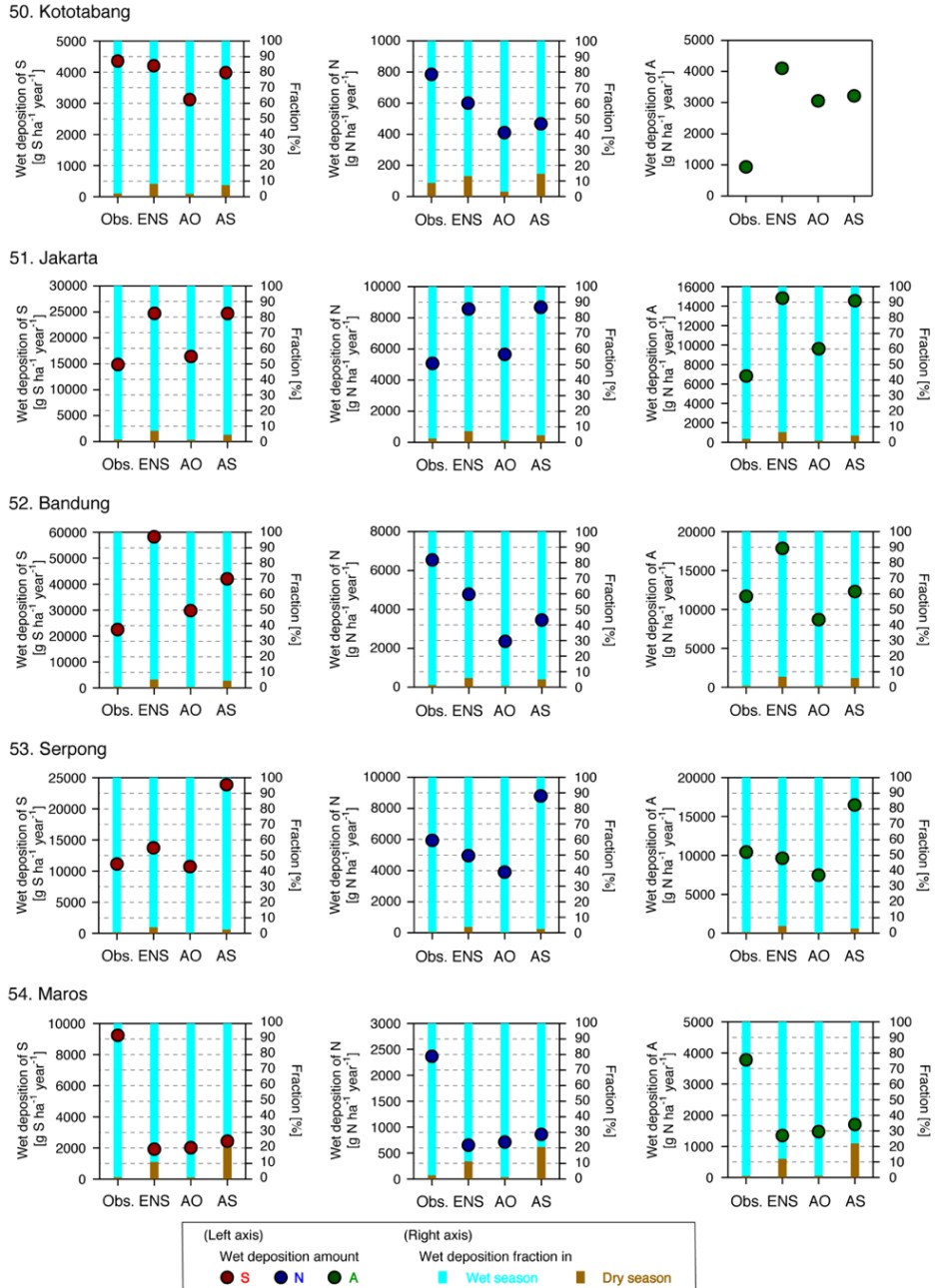

**Figure 16: Observed and simulated annual accumulated wet deposition amounts of S, N, and A, and the fraction of wet deposition during the wet and dry seasons at five sites in Indonesia. The annual accumulated wet deposition amount is based on the months for which wet deposition observations were available (see Fig. 8).**

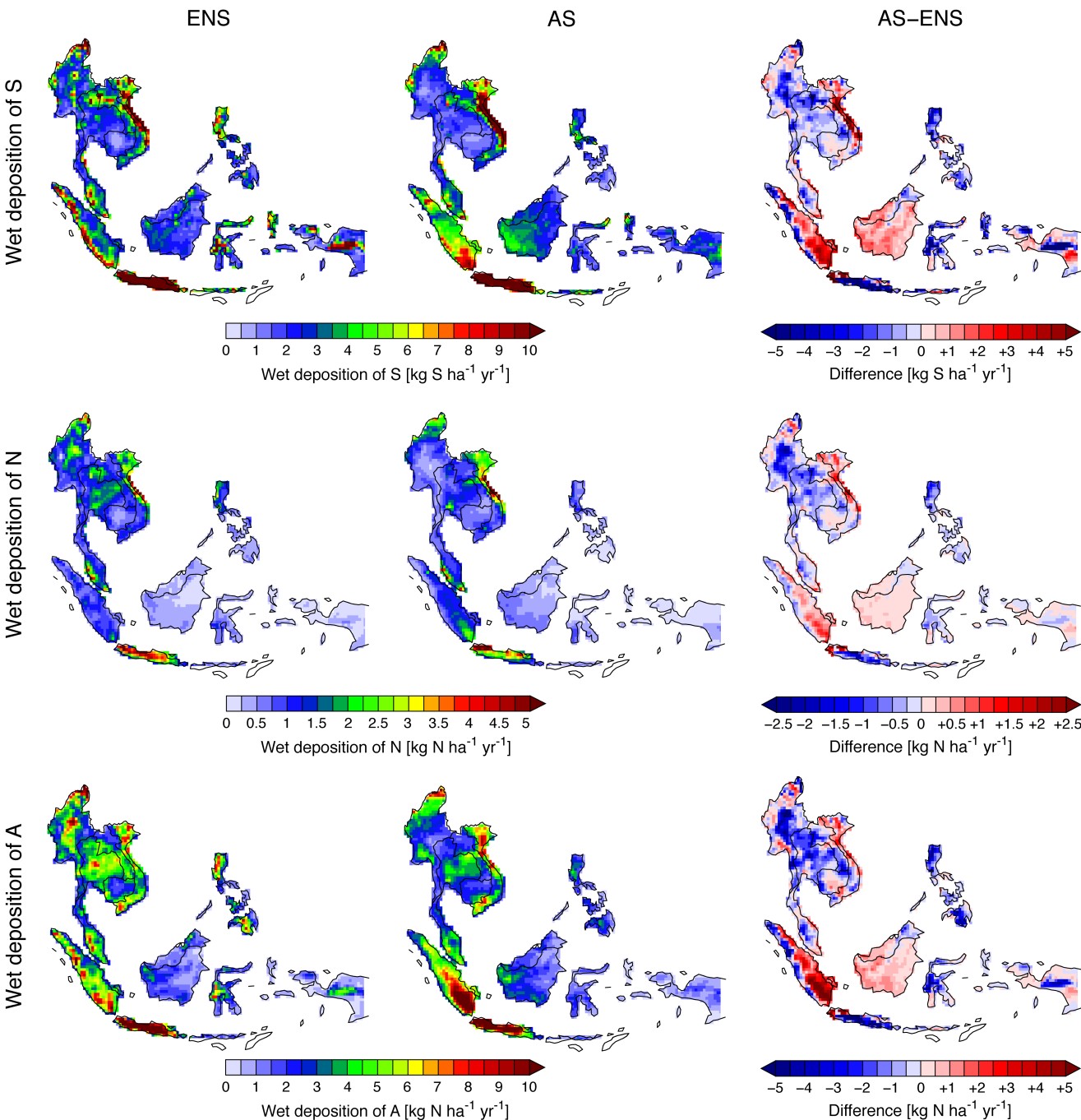

**Figure 17: Maps of the annual accumulated wet deposition of (top) S, (center) N, and (bottom) A calculated by (left) ENS, (middle) AS, and (right) the difference between AS and ENS. Note that the color scale is different for the wet deposition of N. Some locations around the Suva Sea (south of Flores Island, Sumba Island, and Timor Island) and the east of New Guinea Island shown in white are outside of the modeling domain.**

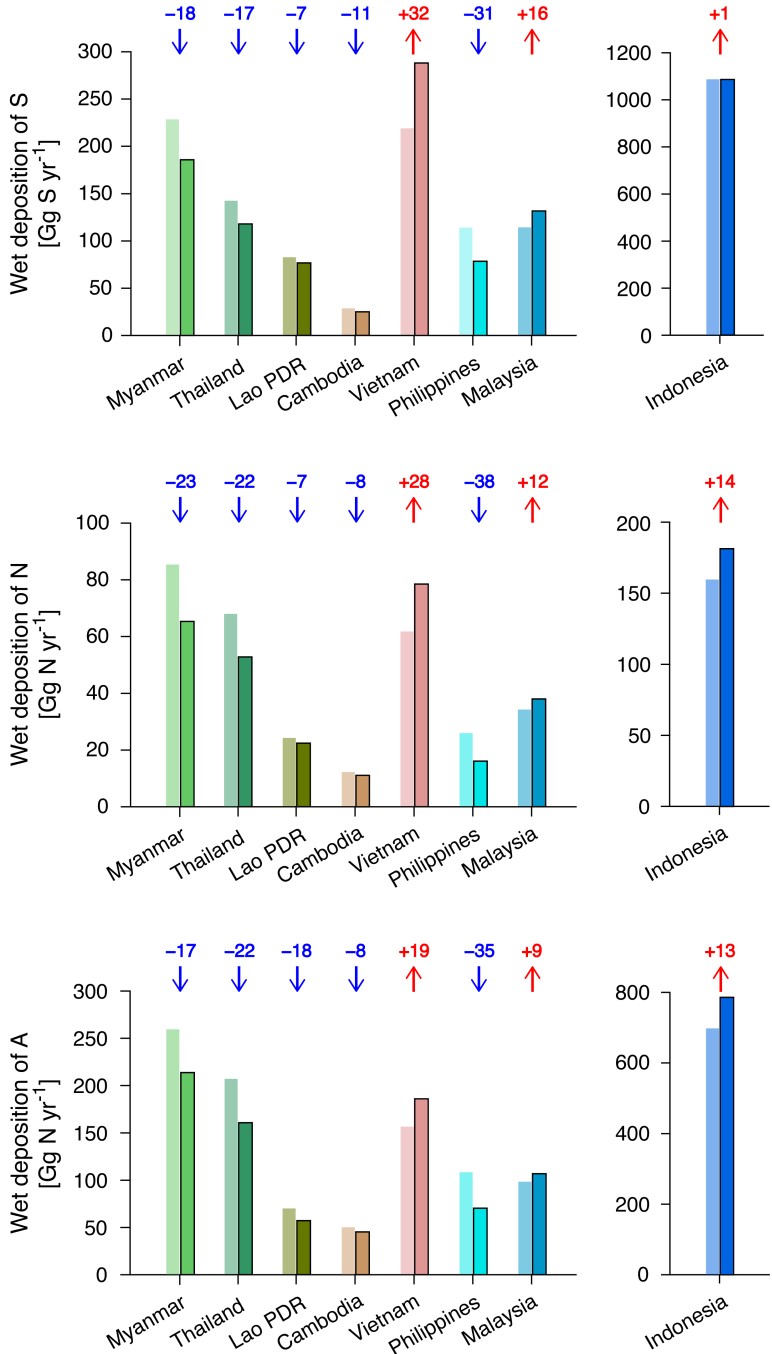

**Figure 18: Simulated annual accumulated wet deposition amounts of (top) S, (center) N, and (bottom) A over Southeast Asian countries calculated by ENS (light bars without outlines) and AS (dark bars outlined in black). Blue numbers with down-facing arrows indicate downward revision by AS and red numbers with upward-facing arrows indicate upward revision by AS.**

**Table 1. Descriptions of the models used in this acid deposition study.**

| No. | M1 | M2 | M4 | M5 | M6 | M11 | M12 |
|---|---|---|---|---|---|---|---|
| Model (version) | CMAQ (5.0.2) | CMAQ (5.0.2) | CMAQ (4.7.1) | CMAQ (4.7.1) | CMAQ (4.7.1) | NAQPMS | NHM Chem |
| Horizontal advection[a] | Yamo | Yamo | PPM | PPM | Yamo | WA | WA |
| Vertical advection[a] | PPM | PPM | PPM | PPM | Yamo | WA | WA |
| Horizontal diffusion[b] | multiscale | multiscale | multiscale | multiscale | multiscale | BD | multiscale |
| Vertical diffusion[b] | ACM2 | ACM2 | ACM2 (inline) | ACM2 | ACM2 (inline) | BD | MYJ |
| Gas-phase chemistry[c] | SAPRC-99 | SAPRC-99 | SAPRC-99 | SAPRC-99 | SAPRC-99 | CBMZ | SAPRC-99 |
| Aerosol chemistry[d] | AERO6 | AERO6 | AERO5 | AERO5 | AERO5 | Li | Kajino |
| Thermodynamics[e] | version 2.1 | version 2.1 | version 1.7 | version 1.7 | version 1.7 | version 1.7 | version 2.1 |
| Dry deposition[f] | M3DRY | M3DRY | M3DRY | M3DRY | M3DRY | Wesely | Kajino |
| Surface layer height | 58 m | 58 m | 58 m | 58 m | 58 m | 48 m | 27 m |
| Wet deposition[g] | Foley | Foley | Foley | Foley | Foley | Ge | Kajino |
| Boundary condition[h] | GEOS-Chem | Default | CHASER | CHASER | CHASER | CHASER | CHASER |

a: References for the advection scheme are as follows: Yamo: Yamartino, 1993; PPM: Piecewise Parabolic Method (Colella and Woodward, 1984); WA: Walcek and Aleksic, 1998.

b: References for the diffusion scheme are as follows: ACM2: Asymmetric Convective Model version 2 (Pleim, 2007a,b); BD: Byun and Dennis, 1995; multiscale: Byun and Schere, 2006; MYJ: Janjic, 1994.

c: References for the gas-phase chemistry are as follows: CBMZ: Zaveri and Peters, 1999; SAPRC-99: Carter, 2000.

d: References for the aerosol chemistry are as follows: AERO5: Foley et al., 2010; AERO6: Appel et al., 2013; Kajino: Kajino et al., 2019a; Li: Li et al., 2011.

e: On thermodynamics. All models use ISORROPIA but different versions, namely version 1.7 (Nenes et al., 1998) or version 2.1 (Fountoukis and Nenes, 2008).

f: References for the dry deposition scheme are as follows: M3DRY: Pleim et al., 2001; Kajino: Kajino et al., 2019a; Wesely: Wesely, 1989.

g: References for the wet deposition scheme are as follows: Foley: Foley et al., 2010; Ge: Ge et al., 2014; Kajino: Kajino et al., 2019a.

h: References for the boundary condition are as follows: CHASER: Sudo et al., 2002a,b; GEOS-Chem: Bey et al., 2001. Note that model M2 adopted the default boundary condition in the Community Multiscale Air Quality (CMAQ) modeling system.

15    **Table 2: Information of 25 Acid Deposition Monitoring Network in East Asia (EANET) observation sites located in Southeast Asia.**

| Site no. | Country | Name | Latitude (°) | Longitude (°E) | Altitude (m a.s.l.) | Sampling interval | Classification |
|---|---|---|---|---|---|---|---|
| 30 | Myanmar | Yangon | 16.50 | 96.12 | 22 | Daily | Urban |
| 31 | Thailand | Bangkok | 13.77 | 100.53 | 2 | Daily | Urban |
| 32 | | Samutprakarn | 13.73 | 100.57 | 2 | Daily | Urban |
| 33 | | Pathumthani | 14.03 | 100.77 | 2 | Daily | Rural |
| 34 | | Khanchanaburi | 14.77 | 98.58 | 170 | Daily | Remote |
| 35 | | Nakhon Ratchasima | 14.45 | 101.88 | 418 | Daily | Rural |
| 36 | | Chiang Mai | 18.77 | 98.93 | 350 | Daily | Rural |
| 37 | Lao PDR | Vientiane | 17.00 | 102.00 | 177 | Daily | Urban |
| 38 | Cambodia | Phnom Penh | 11.55 | 104.83 | 10 | Weekly | Urban |
| 39 | Vietnam | Da Nang | 16.04 | 108.21 | 60 | 10 days | Urban |
| 40 | | Hanoi | 21.02 | 105.85 | 5 | Weekly | Urban |
| 41 | | Hoa Binh | 20.82 | 105.33 | 23 | Weekly | Rural |
| 42 | | Cuc Phuong | 20.25 | 105.72 | 155 | 10 days | Remote |
| 43 | Philippines | Metro Manila | 14,63 | 121.07 | 54 | Weekly | Urban |
| 44 | | Los Baños | 14.18 | 121.25 | 35 | Weekly | Rural |
| 45 | | Mt. Sto. Tomas | 16.42 | 120.60 | 1500 | Weekly | Rural |
| 46 | Malaysia | Petaling Jaya | 3.10 | 101.65 | 87 | Weekly | Urban |
| 47 | | Tanah Rata | 4.47 | 101.38 | 1470 | Weekly | Remote |
| 48 | | Kuching | 1.48 | 110.47 | 22 | Weekly | Urban |
| 49 | | Danum Valley | 4.98 | 117.85 | 427 | Weekly | Remote |
| 50 | Indonesia | Kototabang | −0.20 | 100.32 | 864 | Weekly | Remote |
| 51 | | Jakarta | −6.18 | 106.83 | 7 | Weekly | Urban |
| 52 | | Bandung | −6.90 | 107.58 | 743 | Daily | Urban |
| 53 | | Serpong | −6.25 | 106.57 | 46 | Daily | Rural |
| 54 | | Maros | −4.92 | 119.57 | 11 | Weekly | Rural |

16    Note: Site nos. are unified with the overview paper of Itahashi et al. (2020).

18    **Table 3: Statistical analysis of the model performance for Yangon, Myanmar.**

| | Wet deposition of S | | | Wet deposition of N | | | Wet deposition of A | | |
|---|---|---|---|---|---|---|---|---|---|
| | ENS | AO | AS | ENS | AO | AS | ENS | AO | AS |
| N | | 12 | | | 12 | | | 12 | |
| mean (observation) | | 275.0 | | | 132.0 | | | 388.1 | |
| mean (model) | 62.2 | 96.2 | 102.8 | 31.2 | 44.3 | 48.2 | 111.1 | 168.7 | 182.2 |
| R | 0.81 | 0.77 | 0.65 | 0.74 | 0.79 | 0.69 | 0.87 | **0.95** | 0.83 |
| NMB [%] | −77.4 | **−65.0** | **−62.6** | −76.4 | **−66.4** | **−63.5** | −71.4 | **−56.4** | **−53.1** |
| NME [%] | +84.2 | **+72.7** | **+72.0** | +86.4 | **+72.2** | **+70.8** | +82.1 | **+57.6** | **+53.2** |
| FAC2 [%] | 8.3 | **33.3** | 8.3 | 8.3 | **41.7** | **33.3** | 0.0 | **50.0** | **41.7** |
| FAC3 [%] | 16.7 | **50.0** | **50.0** | 16.7 | **58.3** | **41.7** | 8.3 | **75.0** | **66.7** |
| FAC5 [%] | 33.3 | **91.7** | **58.3** | 25.0 | **91.7** | **58.3** | 33.3 | **91.7** | **66.7** |

19    Note: Units are g S ha$^{-1}$ month$^{-1}$ for the wet deposition of S, and g N ha$^{-1}$ month$^{-1}$ for the wet depositions of N and A. Improvements in the
20    statistical score with AO and AS compared to ENS are shown by bold font.

22   **Table 4: Statistical analysis of the model performance for six sites in Thailand.**

| | Wet deposition of S | | | Wet deposition of N | | | Wet deposition of A | | |
|---|---|---|---|---|---|---|---|---|---|
| | ENS | AO | AS | ENS | AO | AS | ENS | AO | AS |
| N | | 67 | | | 63 | | | 63 | |
| mean (observation) | | 384.8 | | | 309.4 | | | 505.8 | |
| mean (model) | 262.4 | 216.3 | 202.9 | 155.8 | 160.0 | 140.7 | 385.4 | 342.5 | 304.5 |
| R | −0.01 | **0.71** | **0.61** | 0.47 | **0.77** | **0.77** | 0.21 | **0.85** | **0.78** |
| NMB [%] | −31.8 | −43.8 | −47.3 | −49.6 | **−48.3** | −54.5 | −23.8 | −32.3 | −40.0 |
| NME [%] | +86.9 | **+53.6** | **+64.3** | +71.3 | **+53.6** | **+59.7** | +70.8 | **+40.1** | **+48.3** |
| FAC2 [%] | 31.3 | **52.2** | **35.8** | 39.7 | **44.4** | **41.3** | 46.0 | **63.5** | **49.2** |
| FAC3 [%] | 59.7 | **77.6** | **62.7** | 63.5 | **66.7** | 55.6 | 65.1 | **82.5** | 65.1 |
| FAC5 [%] | 77.6 | **92.5** | **79.1** | 81.0 | **84.1** | 71.4 | 85.7 | **95.2** | 76.2 |

23   Note: Units are g S ha$^{-1}$ month$^{-1}$ for the wet deposition of S, and g N ha$^{-1}$ month$^{-1}$ for the wet depositions of N and A. Improvements in the
24   statistical score with AO and AS compared with ENS are shown by bold font.

**Table 5: Statistical analysis of the model performance for Phnom Penh, Cambodia.**

| | Wet deposition of S | | | Wet deposition of N | | | Wet deposition of A | | |
|---|---|---|---|---|---|---|---|---|---|
| | ENS | AO | AS | ENS | AO | AS | ENS | AO | AS |
| N | | 12 | | | 12 | | | 12 | |
| mean (observation) | | 363.7 | | | 180.7 | | | 488.6 | |
| mean (model) | 101.1 | 187.5 | 158.8 | 39.4 | 79.4 | 71.4 | 181.7 | 369.3 | 313.0 |
| R | 0.05 | **0.56** | **0.31** | 0.28 | **0.51** | **0.30** | 0.34 | **0.84** | **0.61** |
| NMB [%] | −72.2 | **−48.4** | **−56.3** | −78.2 | **−56.0** | **−60.5** | −62.8 | **−24.4** | **−35.9** |
| NME [%] | +78.8 | **+57.0** | **+66.2** | +80.9 | **+60.1** | **+66.8** | +69.9 | **+31.0** | **+50.8** |
| FAC2 [%] | 25.0 | **58.3** | **33.3** | 8.3 | **33.3** | **33.3** | 25.0 | **91.7** | **58.3** |
| FAC3 [%] | 25.0 | **66.7** | **66.7** | 25.0 | **66.7** | **41.7** | 50.0 | **91.7** | **91.7** |
| FAC5 [%] | 58.3 | **91.7** | **83.3** | 41.7 | **83.3** | **75.0** | 75.0 | **100.0** | **91.7** |

Note: Units are g S ha$^{-1}$ month$^{-1}$ for the wet deposition of S, and g N ha$^{-1}$ month$^{-1}$ for the wet depositions of N and A. Improvements in the

statistical score with AO and AS compared with ENS are shown by bold font.

**Table 6: Statistical analysis of the model performance for four sites in Vietnam.**

| | Wet deposition of S | | | Wet deposition of N | | | Wet deposition of A | | |
|---|---|---|---|---|---|---|---|---|---|
| | ENS | AO | AS | ENS | AO | AS | ENS | AO | AS |
| N | | 43 | | | 41 | | | 55 | |
| mean (observation) | | 1060.5 | | | 321.5 | | | 486.0 | |
| mean (model) | 673.5 | 700.3 | 756.1 | 215.4 | 249.0 | 274.9 | 559.1 | 579.6 | 590.9 |
| R | 0.63 | **0.67** | **0.67** | 0.47 | **0.59** | **0.48** | 0.46 | **0.57** | **0.60** |
| NMB [%] | −36.5 | **−34.0** | **−19.3** | −33.0 | **−23.9** | **−14.5** | +15.0 | +19.2 | +22.2 |
| NME [%] | +48.2 | **+46.6** | **+42.6** | +55.8 | **+47.6** | **+54.7** | +57.1 | **+52.6** | **+56.2** |
| FAC2 [%] | 53.5 | 51.2 | **60.5** | 48.8 | **56.1** | 65.9 | 41.8 | **41.8** | **41.8** |
| FAC3 [%] | 72.1 | **76.7** | **81.4** | 80.5 | 78.0 | 75.6 | 52.7 | **61.8** | **60.5** |
| FAC5 [%] | 93.0 | **95.3** | 83.7 | 87.8 | 85.4 | 82.9 | 56.4 | **65.5** | 56.4 |

Note: Units are g S ha$^{-1}$ month$^{-1}$ for the wet deposition of S, and g N ha$^{-1}$ month$^{-1}$ for the wet depositions of N and A. Improvements in the
statistical score with AO and AS compared with ENS are shown by bold font.

**Table 7: Statistical analysis of the model performance for three sites in the Philippines.**

| | Wet deposition of S | | | Wet deposition of N | | | Wet deposition of A | | |
|---|---|---|---|---|---|---|---|---|---|
| | ENS | AO | AS | ENS | AO | AS | ENS | AO | AS |
| N | | 28 | | | 20 | | | 22 | |
| mean (observation) | | 594.0 | | | 275.8 | | | 400.4 | |
| mean (model) | 661.5 | 216.3 | 202.9 | 214.6 | 137.9 | 87.4 | 538.2 | 336.7 | 217.7 |
| R | 0.79 | 0.78 | 0.74 | 0.25 | 0.23 | **0.39** | 0.35 | 0.26 | **0.45** |
| NMB [%] | +11.4 | −28.6 | −46.7 | −22.2 | −50.0 | −68.3 | +34.4 | **−15.9** | −45.6 |
| NME [%] | +58.0 | **+45.1** | **+55.3** | +75.1 | **+74.8** | **+70.2** | +123.6 | **+102.9** | **+74.4** |
| FAC2 [%] | 53.6 | **71.4** | **60.7** | 55.0 | 45.0 | 40.0 | 22.7 | 13.6 | **40.9** |
| FAC3 [%] | 60.7 | **89.3** | **75.0** | 60.0 | 50.0 | **60.0** | 50.0 | 27.3 | **54.5** |
| FAC5 [%] | 78.6 | **96.4** | **82.1** | 65.0 | 55.0 | **70.0** | 59.1 | **59.1** | **72.7** |

Note: Units are g S ha$^{-1}$ month$^{-1}$ for the wet deposition of S, and g N ha$^{-1}$ month$^{-1}$ for the wet depositions of N and A. Improvements in the
statistical score with AO and AS compared with ENS are shown by bold font.

**Table 8: Statistical analysis of model performance for four sites in Malaysia.**

| | Wet deposition of S | | | Wet deposition of N | | | Wet deposition of A | | |
|---|---|---|---|---|---|---|---|---|---|
| | ENS | AO | AS | ENS | AO | AS | ENS | AO | AS |
| N | | 37 | | | 36 | | | 36 | |
| mean (observation) | | 709.2 | | | 755.8 | | | 131.5 | |
| mean (model) | 532.6 | 444.9 | 410.3 | 189.3 | 149.7 | 134.2 | 488.1 | 404.0 | 363.3 |
| R | 0.43 | **0.60** | **0.38** | 0.59 | **0.69** | 0.48 | 0.08 | **0.27** | **0.29** |
| NMB [%] | −24.9 | −36.8 | −42.1 | −74.9 | −80.2 | −82.2 | +271.2 | **+207.2** | **+176.3** |
| NME [%] | +69.7 | **+53.6** | **+54.1** | +83.7 | **+83.4** | **+79.6** | +284.5 | **+210.7** | **+180.6** |
| FAC2 [%] | 32.4 | **62.2** | **45.9** | 22.2 | **25.0** | **30.6** | 19.4 | 13.9 | **27.8** |
| FAC3 [%] | 73.0 | **83.8** | 70.3 | 33.3 | **36.1** | 33.3 | 33.3 | **41.7** | **41.7** |
| FAC5 [%] | 94.6 | **100.0** | 91.9 | 50.0 | **61.1** | **61.1** | 63.9 | **72.2** | **80.6** |

Note: Units are g S ha$^{-1}$ month$^{-1}$ for the wet deposition of S, and g N ha$^{-1}$ month$^{-1}$ for the wet depositions of N and A. Improvements in the
statistical score with AO and AS compared with ENS are shown by bold font.

**Table 9: Statistical analysis of model performance for five sites in Indonesia.**

| | Wet deposition of S | | | Wet deposition of N | | | Wet deposition of A | | |
|---|---|---|---|---|---|---|---|---|---|
| | ENS | AO | AS | ENS | AO | AS | ENS | AO | AS |
| N | | 59 | | | 57 | | | 58 | |
| mean (observation) | | 1052.5 | | | 363.2 | | | 580.5 | |
| mean (model) | 1743.1 | 1052.4 | 1644.9 | 343.3 | 228.9 | 390.4 | 823.8 | 466.9 | 856.3 |
| R | 0.68 | **0.89** | **0.71** | 0.56 | **0.70** | **0.57** | 0.47 | **0.45** | **0.50** |
| NMB [%] | +65.6 | **0.0** | **+56.3** | −5.5 | −37.0 | +7.5 | +41.9 | **−2.3** | **+27.5** |
| NME [%] | +100.2 | **+37.7** | **+86.1** | +56.3 | **+49.2** | +63.3 | +79.3 | **+61.6** | **+43.9** |
| FAC2 [%] | 52.5 | **76.3** | 42.4 | 59.6 | 49.1 | 54.4 | 43.1 | 41.4 | **44.8** |
| FAC3 [%] | 71.2 | **83.1** | 69.5 | 73.7 | 71.9 | **73.7** | 50.0 | **53.4** | **58.6** |
| FAC5 [%] | 79.7 | **91.5** | **83.1** | 80.7 | **87.7** | **89.5** | 58.6 | **60.3** | **70.7** |

Note: Units are g S ha$^{-1}$ month$^{-1}$ for the wet deposition of S, and g N ha$^{-1}$ month$^{-1}$ for the wet depositions of N and A. Improvements in the
statistical score with AO and AS compared with ENS are shown by bold font.