# Peer review of "Insights into seasonal variation of wet deposition over Southeast Asia via precipitation adjustment from the findings of MICS-Asia III"

_Atmospheric Chemistry and Physics, 2020_

## Referee Comment (RC1) · Anonymous Referee #1 · 28 Jan 2021

General comments

This study presents an interesting analysis of modeled wet deposition in Southeast Asian and proposes an approach to improve wet deposition estimates using two precipitation datasets. The manuscript is well written and clearly structured, the figures and tables are of high quality, and the analysis methods and underlying model simulations are sound. My comments mainly focus on the interpretation of the results which I feel could be expanded and improved upon prior to publication.

Specific comments

Page 4, lines 5 – 12: While the details of the model configurations have been published

before, I think it would be useful to add a table summarizing some key aspects (model resolution, chemical mechanism, aerosol scheme, wet and dry deposition scheme) of the models used in this study. This would help in the interpretation of model performance, model-to-model variability, ensemble construction, and needs for future model development.

Page 4, line 17: My interpretation of Solazzo et al. (2012) is that ensemble approaches other than averaging over all available models with equal weights performed better than the ensemble mean. Were such other ensemble approaches explored in MICS-Asia Phase 3, especially given that 5 of the 7 models used in this study were CMAQ and may have had greater similarity than the other two models?

Page 5, Lines 35 – 37: please clarify how the aggregation of weekly or ten day observations to monthly values was handled when a sampling period spanned across two months, and whether the same approach was used to calculate monthly model values in these instances.

Page 7, line 96: can you please state what criteria were used to define an "acceptable level"?

Page 8, lines 48 – 50: I think it would be really important to expand this discussion and provide a motivation why the precipitation-adjustment approach was viewed as the most appropriate and effective avenue for improving model performance. I agree that biases in precipitation are critical to consider when evaluating and trying to improve modeled wet deposition, but I would also like to see a discussion (maybe a summary of Itahashi et al., 2020) of other potential drivers of model biases, not just in wet deposition but also the actual concentrations of the compounds analyzed in this study. For example, what is known from other MICS-Asia work about whether some of the wet deposition performance issues might be caused by errors in the emissions of oxidized and reduced nitrogen and uncertainties in the representation of atmospheric chemistry in the models? The model-to-model variability indicated by the whiskers in Figures 1

– 7 is substantial and is not driven by precipitation since all models used the same WRF fields, yet no mention of this variability is made in sections 3.1.1 – 3.1.7 and its implications for improving the modeled wet deposition fields are not discussed in this section. An acknowledgment that precipitation errors are likely not the sole drivers of error in wet deposition should also be added to the abstract (page 2, lines, 39-40) and conclusions (page 13, lines 04-05)

Page 9, lines 73 – 80: I like the idea of including both the EANET observations and TRMM satellite data in this analysis. What I find interesting and would like to see discussed in more detail is the fact that the squared correlation coefficient between EANET and TRMM data is only about 0.5, i.e. only about 50% of the precipitation variability seen in the point observations is captured by the satellite product. To the extent that these point observations (both for precipitation and wet deposition) are used to evaluate the model, what does this relatively low correlation say about the spatial scales represented by the three different datasets (observations, model simulations, and TRMM), their commensurability, and inferences drawn from differences between them?

Page 9, lines 82-83: I am not sure I follow this argument. If the EANET observations are the ground truth and TRMM has a correlation of only 0.7 with them, why would one consider the TRMM-based AS adjustment AS over AO? I realize that the conceptual benefit is that AS can be applied across the entire domain while AO is limited to specific stations as discussed at the beginning of section 4.3, but this is not the argument I'm reading here.

Page 13, lines 77 – 83, discussion of Figure 17. I recommend avoiding the terms "overestimation" and "underestimation" when discussing the spatial patterns of wet deposition rather than precipitation results. I suggest to instead use terms like "higher values" and "lower values". Aside from the EANET station analysis already presented in Figures 10 – 16, no observational data is available to judge whether the AS deposition patterns are higher or lower than reality.

Page 13, lines 89 – 90: Since no actual country-wide observations are available aside from the EANET point measurements, I recommend avoiding terms like "reproducibility" and "accurate estimation" when discussing Figure 18.

Technical corrections

Page 2, Line 40: change "scaling" to "scaled"

Page 2, Line 42: suggesting changing "spatio-and-temporal" to "spatio-temporal"

Page 3, Line 97: change "The participating model was requested" to "The participating models were requested"

Page 6, line 79: suggest changing "calculated" to "modeled"

Page 13, lines 01 – 02: please double check the wording of this sentence, it reads like a contradiction to me "Generally, the ensemble model can capture the observed wet deposition; however, the models failed to capture the wet deposition, even the ensemble mean . . ."

---

## Referee Comment (RC2) · Anonymous Referee #2 · 7 Feb 2021

Overall comments This manuscript is dealing with the wet depositions of S, N, and A in Southeast Asia. In order to estimate their monthly wet deposition amounts, authors analyzed nine air quality modeling outputs with different CTM models, versions, and the configurations, and used the ensemble mean of them to compare the results to the observations. For the better match with the EANET observations, authors introduced the precipitation-weighted wet deposition adjustments based on the observed precipitations at the weather stations and from the satellite measurements. This manuscript is well written, but a few issues should be addressed clearly before the consideration of publication in the journal. Please see the major and specific questions and comments below.

Major comments: 1. The main purpose of the manuscript is not clear. Authors might have focused on the measurements of wet depositions of S, N, and A, and their spatial distributions. On the other hand, it is not certain what the insight of the precipitation weighted adjustments authors want to tell. 2. Authors also need to explain the advantage of the so-called precipitation weighted adjustments of the monthly wet depositions. In my opinion, the wet deposition map in Southeast Asia can be directly developed with the observation data over the region by applying a spatial interpolation. Considering the uncertainties laid on the estimation, it is not sure how the new estimation is reliable or can be applied for the future research. 3. It is presumed that the precipitation amount is the dominant factor to determine the total wet deposition amounts. But the modeled amounts of the wet depositions can vary depending on the airborne concentrations. However, no model validation for the concentrations of air pollutants is available in the manuscript. 4. Still the proposed adjustment method for the wet deposition would be useful. However, the limitation and cautions in the use should be discussed in detail.

Minor comments: Line 114-115: In Eq (1), how is the ensemble mean different from the mean of individual models? Authors mentioned here the ensemble mean better matched to the observations, but it seems the mean of individual models is used in Eq. (1).

Line 143: Can authors explain what 'percentages' is meant here more clearly?

Line 151: Are there any approaches to evaluate the airborne concentrations first? Over- or under-predictions of the airborne concentrations may lead to the discrepancy between the observations and simulations.

Line 158: Light precipitation explained in Lines 54-55 might have caused the over-estimation of modeled wet deposition. Have authors evaluated or analyzed the role of rain intensity and the rainfall hours to control the amounts of the modeled wet depositions? In Fig 2, compared to the modeled precipitation amounts, the models over-predicted the wet-deposited amounts more excessively.

Line 164-165: More specifically, is this due to the meteorology model issue or the algorithm in the CTMs?

Line 174: In Fig 3, compared to the monthly variations of precipitations, those of the wet depositions are relatively small. The absolute amount of precipitation plays a role of determining the wet depositions, but it would not be critical.

Line 185: Is the ENS calculated for the one EANET site?

Line 248: That could be one of reasons, but still not sure how the precipitation significantly affects the magnitude of wet depositions. Authors may define 'precipitation' in the manuscript. As I understand, precipitation that affects the wet depositions of air pollutants include the rainfall amount, intensity, frequency, and the duration.

Line 258: There clearly exist under-predictions of precipitations during dry season while over-prediction during wet season. Therefore, authors may apply the adjustment for dry and wet season separately instead of the annual total. For example, precipitations in Thailand during dry season was under-predicted, but those for wet season was over-predicted in the model.

Line 376: Site-specific adjustment factors utilizing the observation data can be applied to revise the wet deposition amounts in the spatial plots. If the main purpose of the precipitation-weighted adjustment of the wet depositions is to derive the realistic data close to the observations, why is a simple method like a spatial interpolation of the observed data not applied in this study? What advantage can we expect from the modeled wet deposition adjustments introduced in this study instead of a simple method?

---

## Author Comment (AC1) · 2 Apr 2021

Response to Comment 1 by Anonymous Reviewer 1

**We thank you for providing helpful and constructive comments and suggestions. We have revised our manuscript accordingly. We hope that these revisions satisfactorily address all the points you have raised. Our point-by-point responses are provided below, and revisions are indicated in blue in the revised manuscript.**

General comments

This study presents an interesting analysis of modeled wet deposition in Southeast Asian and proposes an approach to improve wet deposition estimates using two precipitation datasets. The manuscript is well written and clearly structured, the figures and tables are of high quality, and the analysis methods and underlying model simulations are sound. My comments mainly focus on the interpretation of the results which I feel could be expanded and improved upon prior to publication.

**Reply:**

**We appreciate this positive comment. To address your concerns related to the interpretation of the results, we have revised the manuscript according to your comments.**

Specific comments

Page 4, lines 5 – 12: While the details of the model configurations have been published before, I think it would be useful to add a table summarizing some key aspects (model resolution, chemical mechanism, aerosol scheme, wet and dry deposition scheme) of the models used in this study. This would help in the interpretation of model performance, model-to-model variability, ensemble construction, and needs for future model development.

**Reply:**

**Thank you for this helpful comment. We have included a new Table 1 to present descriptions (horizontal and vertical advection/diffusion, gas and aerosol chemistry, dry and wet deposition schemes, and boundary condition) of the seven models used in this study. We also added the relevant references in this new Table 1. As the introduction to this new**

**table, we have added the following sentences in Section 2.1 on P4, L16.**

**"In this study, seven models (M1, M2, M4, M5, M6, M11, and M12) that using the same meteorological fields simulated by the Weather Research and Forecasting (WRF) model version 3.4.1 (Skamarock et al., 2008) over the unified modeling domain were selected. The unified modeling domain covered the whole of Asia with a horizontal grid resolution of 45 km and 40 vertical layers from the surface up to 10 hPa. Descriptions of the seven models are listed in Table 1"**

Page 4, line 17: My interpretation of Solazzo et al. (2012) is that ensemble approaches other than averaging over all available models with equal weights performed better than the ensemble mean. Were such other ensemble approaches explored in MICS-Asia Phase 3, especially given that 5 of the 7 models used in this study were CMAQ and may have had greater similarity than the other two models?

**Reply:**

**Thank you for this insightful comment. In our overview paper (Itahashi et al., 2020), we tried to apply a simple ensemble average and weighted ensemble approach. To clarify this point, we have added the following sentences in Section 2.2 on P5, L10.**

**"Solazzo et al. (2012) proposed a method to produce a better ensemble. In the deposition analysis of MICS-Asia Phase III, a simple ensemble and a weighted ensemble were performed using the correlation coefficient (R) between the modeled and observed wet deposition (Itahashi et al., 2020). It was found that R was always improved by the weighted ensemble; however, biases can be worse in a weighted ensemble for some cases."**

Page 5, Lines 35 – 37: please clarify how the aggregation of weekly or ten day observations to monthly values was handled when a sampling period spanned across two months, and whether the same approach was used to calculate monthly model values in these instances.

**Reply:**

**We appreciate this question. To explain the methodology used for monthly averaging, we have added the following sentences on P5, L27.**

**"The monthly accumulated wet deposition at each site were used for the model evaluation. For weekly or 10-day observational data, the central observation day was regarded to represent the corresponding month, and then the monthly accumulated wet deposition was calculated."**

**Additionally, to clarify this information, the sampling intervals at each observation site have been added in Table 2.**

**For models, the monthly average was simply calculated based on the date. We have added this point on P5, L29.**

**"Meanwhile, the model results were simply calculated from the calendar date."**

Page 7, line 96: can you please state what criteria were used to define an "acceptable level"?

**Reply:**

**Because there are no criteria to define this level, we have removed this point in the revised manuscript.**

Page 8, lines 48 – 50: I think it would be really important to expand this discussion and provide a motivation why the precipitation-adjustment approach was viewed as the most appropriate and effective avenue for improving model performance. I agree that biases in precipitation are critical to consider when evaluating and trying to improve modeled wet deposition, but I would also like to see a discussion (maybe a summary of Itahashi et al., 2020) of other potential drivers of model biases, not just in wet deposition but also the actual concentrations of the compounds analyzed in this study. For example, what is known from other MICS-Asia work about whether some of the wet deposition performance issues might be caused by errors in the emissions of oxidized and reduced nitrogen and uncertainties in the representation of atmospheric chemistry in the models? The model-to-model variability indicated by the whiskers in Figures 1–7 is substantial and is not driven by precipitation since all models used the same WRF fields, yet no mention of this variability is made in sections 3.1.1 – 3.1.7 and its implications for improving the modeled wet deposition fields are not discussed in this section. An acknowledgment that precipitation errors are likely not the sole drivers of error in wet deposition should also be added to the abstract (page 2, lines, 39-40) and conclusions (page 13, lines

04-05).

**Reply:**

**We are grateful for this insightful comment on our approach. As pointed out, we omitted the discussion of the ambient concentration itself, so we carefully revised our manuscript to mention this point. In terms of the model evaluation for ambient concentrations, our companion paper (Chen et al., 2019) overviewed the modeling performance of MICS-Asia Phase III. Moreover, the variation among the seven models has now been added in each discussion section. The additional discussions of the inter-model variability are as follows:**

**For Myanmar, in Section 3.1.1 on P6, L21:**

[revised manuscript text omitted]

This point was also added in the abstract to explain why the adjustment method was applied in this study. The additional sentences are as follows on P2, L4:

"Considering the model performance for ambient aerosol concentrations over Southeast Asia, this failure of models is considered to be related to the difficulty in capturing the precipitation in Southeast Asia, especially during the dry and wet seasons. Generally, meteorological field overestimated the precipitation during the dry season, which leads to the overestimation of wet deposition during this season. To overcome this, a precipitation-adjusted approach that scaled the modeled precipitation to the observed value was applied, and it was demonstrated that the model performance was improved."

Page 9, lines 73 – 80: I like the idea of including both the EANET observations and TRMM satellite data in this analysis. What I find interesting and would like to see discussed in more detail is the fact that the squared correlation coefficient between EANET and TRMM data is only about 0.5, i.e. only about 50% of the precipitation variability seen in the point observations is captured by the satellite product. To the extent that these point observations (both for precipitation and wet deposition) are used to evaluate the model, what does this relatively low correlation say about the spatial scales represented by the three different datasets (observations, model simulations, and TRMM), their commensurability, and inferences drawn from differences between them?

Reply:

We are grateful for this helpful comment on the differences in precipitation. We have added the following point regarding the requirements for future studies on P11, L26.

"It should be noted that even though satellite and ground-based observations showed differences in the precipitation amount, this result indicates that further consideration of the how well precipitation is represented by the spatial resolution (broader observation by satellites and point-specific observations using ground-based monitoring) is important. Accordingly, the effect of the modeling spatial resolution on the simulated precipitation should be considered in future studies."

Page 9, lines 82-83: I am not sure I follow this argument. If the EANET observations are the ground truth and TRMM has a correlation of only 0.7 with them, why would one consider the TRMM-based AS adjustment AS over AO? I realize that the conceptual benefit is that AS can be applied across the entire domain while AO is limited to specific stations as discussed at the beginning of section 4.3, but this is not the argument I'm reading here.

Reply:

The simulated wet deposition based on the original simulated precipitation shown in Figure 9 (left) has been evaluated in Figures 2–8. Based on the precipitation-adjustment via Eq. (5), the simulated wet deposition is revised. Satellite measurement compared to EANET ground observations shown in Figure 9 (right) obtained a more accurate precipitation amount rather than the simulation. The better estimation of precipitation itself will lead to a better simulation of wet deposition. In this sentence, we did not consider the better performance of AS compared to AO; rather, we only discussed the model improvements relative to the original simulation results.

To avoid misunderstanding, this point has been revised as follows on P11, L25.

"From this result, it is expected that precipitation-adjustment based on satellite measurements also has the potential to improve the original simulation of wet deposition."

Page 13, lines 77 – 83, discussion of Figure 17. I recommend avoiding the terms "overestimation" and "underestimation" when discussing the spatial patterns of wet deposition rather than precipitation

results. I suggest to instead use terms like "higher values" and "lower values". Aside from the EANET station analysis already presented in Figures 10 – 16, no observational data is available to judge whether the AS deposition patterns are higher or lower than reality.

**Reply:**

**We agree with this suggestion. We have revised the manuscript to use "higher/lower values" for the discussion of Figure 17.**

Page 13, lines 89 – 90: Since no actual country-wide observations are available aside from the EANET point measurements, I recommend avoiding terms like "reproducibility" and "accurate estimation" when discussing Figure 18.

**Reply:**

**We have removed the use of "reproducibility" from the revised manuscript by deleting the redundant sentence. The expression "accurate estimation" was not used in relation to Figure 18, and we would like to retain the use of this expression elsewhere.**

Technical corrections

Page 2, Line 40: change "scaling" to "scaled"

**Reply:**

**We have corrected this.**

Page 2, Line 42: suggesting changing "spatio-and-temporal" to "spatio-temporal"

**Reply:**

**We have corrected this.**

Page 3, Line 97: change "The participating model was requested" to "The participating models were requested"

**Reply:**

**We have corrected this.**

Page 6, line 79: suggest changing "calculated" to "modeled"

**Reply:**

**We have corrected this.**

Page 13, lines 01 – 02: please double check the wording of this sentence, it reads like a contradiction to me "Generally, the ensemble model can capture the observed wet deposition; however, the models failed to capture the wet deposition, even the ensemble mean …"

**Reply:**

**We appreciate this careful checking. We have revised this sentence as follows in P15, L22.**

**"Generally, the ensemble model could capture the observed wet deposition; however, sometimes failed to capture the wet deposition and obtained low correlations and/or large biases and errors."**

---

## Author Comment (AC2) · 2 Apr 2021

Response to Comment 1 by Anonymous Reviewer 2

**We thank you for providing helpful and constructive comments and suggestions. We have revised our manuscript accordingly. We hope that these revisions satisfactorily address all the points you have raised. Our point-by-point responses are provided below, and revisions are indicated in blue in the revised manuscript.**

Overall comments

This manuscript is dealing with the wet depositions of S, N, and A in Southeast Asia. In order to estimate their monthly wet deposition amounts, authors analyzed nine air quality modeling outputs with different CTM models, versions, and the configurations, and used the ensemble mean of them to compare the results to the observations. For the better match with the EANET observations, authors introduced the precipitation-weighted wet deposition adjustments based on the observed precipitations at the weather stations and from the satellite measurements. This manuscript is well written, but a few issues should be addressed clearly before the consideration of publication in the journal. Please see the major and specific questions and comments below.

**Reply:**

**We appreciate the consideration of our manuscript by the journal. We have fully revised the manuscript to address your concerns listed below.**

Major comments:

1. The main purpose of the manuscript is not clear. Authors might have focused on the measurements of wet depositions of S, N, and A, and their spatial distributions. On the other hand, it is not certain what the insight of the precipitation weighted adjustments authors want to tell.

**Reply:**

**We are grateful for this helpful comment on our approach. In order to clarify the purpose of this study, we added the following sentences in the introduction section on P3, L17.**

**"In an overview paper (Itahashi et al., 2020), we presented the acid deposition status over**

**Asia; however, this presentation was mostly limited to the annual-accumulated status. Over Southeast Asia, which experiences distinct dry and wet seasons, wet deposition varies dramatically between these seasons. Detailed analysis is required to advance our understanding of the wet deposition status over this region, which motivated the present study. Additionally, in Itahashi et al. (2020), we reported the uncertainty of the current model-based estimation of wet deposition and proposed two approaches for improving this estimation, namely model ensemble and precipitation adjustment. The former can modulate the differences between models and the latter can adjust the precipitation amount based on observational data."**

2. Authors also need to explain the advantage of the so-called precipitation weighted adjustments of the monthly wet depositions. In my opinion, the wet deposition map in Southeast Asia can be directly developed with the observation data over the region by applying a spatial interpolation. Considering the uncertainties laid on the estimation, it is not sure how the new estimation is reliable or can be applied for the future research.

**Reply:**

**We are grateful for this suggestion. To answer this question and reinforce the advantage of our approach using CTMs, we have added the following sentences in the introduction section on P3, L27.**

**"The available EANET observation sites are limited over Southeast Asia; therefore, spatial interpolation methods (e.g., Kriging, land use regression) that directly use observational data (Briggs et al., 2000; Ross et al., 2007; Araki et al., 2017) may be difficult to apply. Under the framework of MICS-Asia III, an emission inventory over Asia was developed as MIX emissions (Li et al., 2017), and this is used for input data on CTMs in MICS-Asia III and subsequently conducted model inter-comparison study over Asia. Producing maps of the estimated wet deposition through CTMs can be a reasonable approach to achieve this goal."**

3. It is presumed that the precipitation amount is the dominant factor to determine the total wet deposition amounts. But the modeled amounts of the wet depositions can vary depending on the

airborne concentrations. However, no model validation for the concentrations of air pollutants is available in the manuscript.

**Reply:**

**We are grateful for this insightful comment on our approach. This point regarding the discussion of airborne concentrations was also raised by another reviewer. An analysis of airborne concentration has already been presented in our companion paper of Chen et al. (2019) regarding the outcome of MICS-Asia Phase III. We omitted discussion of this point in the original manuscript; however, in the revised manuscript, we have added the following statement on airborne concentration in Section 4.1 on P10, L10.**

**"The errors in the simulated values of wet deposition are associated with ambient concentration and/or precipitation. Our previous overview paper (Itahashi et al., 2020) presented two approaches for improving the modeling of wet deposition, namely, the ensemble approach and the precipitation-adjusted approach. The former approach was used in this study. In terms of the modeling performance for the ambient concentrations of aerosols of $SO_4^{2-}$, $NO_3^-$, and $NH_4^+$, our companion paper reported better performance over Southeast Asia compared with North and East Asia (Chen et al., 2019). As noted in Section 3, the model generally overestimated precipitation as well as wet deposition during the dry season. Additionally, the model sometimes simulated non-zero precipitation, and consequently non-zero wet deposition, despite the absence of wet deposition due to the absence of precipitation. Based on these findings in MICS-Asia III, the difficulty stemmed from the inaccuracy of the modeled precipitation, which is fundamentally important for simulating the wet deposition. The precipitation-adjustment method is expected to improve model performance."**

4. Still the proposed adjustment method for the wet deposition would be useful. However, the limitation and cautions in the use should be discussed in detail.

**Reply:**

**Thank you very much for this comment on the adjustment approach. We originally addressed the limitation of this method as follows on P11, L1.**

**"Adjustment using shorter time scales is difficult because the modeled precipitation ($P_{model}$**

in Eq. (5)) approaches zero, which leads to unreasonably large values, and vice versa for larger time scales."

Taking into account your comments, as well as the comment from another reviewer, the conclusion section has been revised to repeat the limitation of this method and discuss subjects that are planned to be investigated in MICS-Asia Phase IV. The revised sentences in the conclusion section on P16, L8 are as follows:

"The precipitation-adjustment approach was effective at most sites; however, no improvement was found at other sites. The understanding of the mechanisms of the wet deposition process itself should be further investigated and inter-compared in the future Phase IV. This adjustment approach might be difficult to apply at time scales shorter than one month; therefore, the performance of meteorological models for precipitation simulation should be paid further attention in order to improve the simulation accuracy of wet deposition. Additionally, greater inter-model variation was noted in the Philippines and Indonesia, especially during months with heavy precipitation. To investigate the differences on model wet deposition scheme, such heavy rainy events with finer spatio-temporal resolution should be pursued in the future MICS-Asia Phase IV."

Minor comments:

Line 114-115: In Eq (1), how is the ensemble mean different from the mean of individual models? Authors mentioned here the ensemble mean better matched to the observations, but it seems the mean of individual models is used in Eq. (1).

Reply:

Thank you for your suggestion. This point was also raised by another reviewer. In order to mention the performances of individual models, we have added the following sentences.

For Myanmar, in Section 3.1.1 on P6, L21:

"At the Yangon (No. 30) site, the model variation (shown by whiskers in Fig. 2) was small for the wet depositions of S, N, and A; this indicates that the overestimation during the dry season and underestimation during the wet season was common among all models."

For Thailand, in Section 3.1.2 on P7, L16:

"Large inter-model variability in the modeled wet deposition was found in some months at Khanchanaburi (No. 34). This could be related to the difference in the ambient concentration and the difference in the mechanisms of the wet deposition scheme because all models used the same meteorological field. It should be noted that all models always showed a large wet deposition in February, March, and November, despite the observed zero wet deposition amount in these months (due to the lack of precipitation during the dry season). This suggests that the discrepancy in the simulated precipitation amount could be the cause of the inaccurate simulation of wet deposition"

For Cambodia, in Section 3.1.3 on P8, L1:

"All models commonly underestimated the wet deposition during the wet season."

For Vietnam, in Section 3.1.4 on P8, L12:

"There were large inter-model differences when the precipitation was high. This result suggests that heavy rain events may lead to large inter-model variability in the simulated wet deposition, and the mechanisms should be further investigated. As concluded in the overview paper of Itahashi et al. (2020), this is one of the lessons learned in MICS-Asia Phase III, and this will be addressed as part of the next MICS-Asia"

For the Philippines, in Section 3.1.5 on P8, L26:

"Because of this precipitation overestimation, the ENS also tended to overestimate the wet depositions of S, N, and A. Compared with other countries, the inter-model differences were larger for the sites in the Philippines. Further seeking of model wet deposition schemes focused on this region will be needed."

For Malaysia, in Section 3.1.6 on P9, L15:

"This tendency was common as indicated by the model-to-model variability. At these two sites, observations showed a small wet deposition of N, and the balance between cations and anions should be carefully examined."

and

"The inter-model variability was small; hence, this overestimation could be connected to the overestimation of precipitation."

> For Indonesia, in Section 3.1.7 on P10, L2:
>
> "As was found in the Philippines, the inter-model variation was large, except for Maros (No. 54) , and further study focusing on this region will also be required."
>
> In the conclusion section on P16, L12:
>
> "Additionally, greater inter-model variation was noted in the Philippines and Indonesia, especially during months with heavy precipitation. To investigate the differences on model wet deposition scheme, such heavy rainy events with finer spatio-temporal resolution should be pursued in the future MICS-Asia Phase IV."

Line 143: Can authors explain what 'percentages' is meant here more clearly?

> Reply:
>
> "the percentages" have been revised to "percentage of the total that fell within…" to clarify them.

Line 151: Are there any approaches to evaluate the airborne concentrations first? Over- or under-predictions of the airborne concentrations may lead to the discrepancy between the observations and simulations.

> Reply:
>
> Yes. As we mentioned in our reply to Major Comment 3, we have revised the manuscript to include the modeling performances for airborne concentration in Section 4.1.

Line 158: Light precipitation explained in Lines 54-55 might have caused the overestimation of modeled wet deposition. Have authors evaluated or analyzed the role of rain intensity and the rainfall hours to control the amounts of the modeled wet depositions? In Fig 2, compared to the modeled precipitation amounts, the models overpredicted the wet-deposited amounts more excessively.

> Reply:
>
> No. Under the framework of MICS-Asia Phase III, the submitted wet deposition from each model were monthly accumulated amounts, and it was not possible to obtain wet deposition

at a finer temporal resolution. In the revised manuscript, this point has been included in the conclusion section. Please see our reply to Major Point 4.

Line 164-165: More specifically, is this due to the meteorology model issue or the algorithm in the CTMs?

**Reply:**

**We think this issue is stemmed from both models. In the present study, the WRF meteorological model showed discrepancies compared to the observed result. After we refined the modeled precipitation in the meteorological model, the errors related to CTMs still caused a difference between the simulated and observed wet deposition. We would like to avoid to explicitly mentioning this point.**

Line 174: In Fig 3, compared to the monthly variations of precipitations, those of the wet depositions are relatively small. The absolute amount of precipitation plays a role of determining the wet depositions, but it would not be critical.

**Reply:**

**We appreciate this helpful comment regarding the behavior for Thailand. To address this point, we have added the following sentences in Section 3.1.2 on P7, L9.**

**"Compared to the monthly precipitation pattern, the observed monthly variations of precipitation amount and wet deposition did not show a clear relationship at Khanchanaburi (No. 34), Nakhon Ratchasima (No. 35), or Chiang Mai (No. 36). Over these sites, ambient concentrations might have contributed to the amount of the wet deposition amount."**

**Despite these observed results, the model showed important features, especially at Khanchanaburi (No. 34). We also added the following sentences on P7, L16.**

**"Large inter-model variability in the modeled wet deposition was found in some months at Khanchanaburi (No. 34). This could be related to the difference in the ambient concentration and the difference in the mechanisms of the wet deposition scheme because all models used the same meteorological field. It should be noted that all models always**

**showed a large wet deposition in February, March, and November, despite the observed zero wet deposition amount in these months (due to the lack of precipitation during the dry season). This suggests that the discrepancy in the simulated precipitation amount could be the cause of the inaccurate simulation of wet deposition."**

Line 185: Is the ENS calculated for the one EANET site?

**Reply:**

**Yes. Because only one EANET site was available in Cambodia, we evaluated the model performance by comparing at this one site.**

Line 248: That could be one of reasons, but still not sure how the precipitation significantly affects the magnitude of wet depositions. Authors may define 'precipitation' in the manuscript. As I understand, precipitation that affects the wet depositions of air pollutants include the rainfall amount, intensity, frequency, and the duration.

**Reply:**

**To address this concern, we added the following sentence on P10, L26.**

**"This method involves adjusting the precipitation amount which affects the wet deposition amount on a monthly time scale."**

Line 258: There clearly exist under-predictions of precipitations during dry season while over-prediction during wet season. Therefore, authors may apply the adjustment for dry and wet season separately instead of the annual total. For example, precipitations in Thailand during dry season was under-predicted, but those for wet season was over-predicted in the model.

**Reply:**

**As we stated on L256 of the original manuscript, we applied the precipitation-adjustment on a monthly time-scale and did not perform them as an annual total. We first adjusted the wet deposition on a monthly time-scale and then the annual wet deposition was recalculated from the precipitation-adjusted monthly wet deposition.**

Line 376: Site-specific adjustment factors utilizing the observation data can be applied to revise the wet deposition amounts in the spatial plots. If the main purpose of the precipitation-weighted adjustment of the wet depositions is to derive the realistic data close to the observations, why is a simple method like a spatial interpolation of the observed data not applied in this study? What advantage can we expect from the modeled wet deposition adjustments introduced in this study instead of a simple method?

**Reply:**

**According to Major Point 1, we chose to use CTMs in this study. Please see our reply.**